# Revisiting Neural Scaling Laws in Language and Vision

**Ibrahim Alabdulmohsin**
Google Research, Brain Team
Zürich, Switzerland
ibomohsin@google.com

**Behnam Neyshabur**
Google Research, Blueshift Team
Mountain View, United States
neyshabur@google.com

**Xiaohua Zhai**
Google Research, Brain Team
Zürich, Switzerland
xzhai@google.com

## Abstract

The remarkable progress in deep learning in recent years is largely driven by improvements in scale, where bigger models are trained on larger datasets for longer schedules. To predict the benefit of scale empirically, we argue for a more rigorous methodology based on the extrapolation loss, instead of reporting the best-fitting (interpolating) parameters. We then present a recipe for estimating scaling law parameters reliably from learning curves. We demonstrate that it extrapolates more accurately than previous methods in a wide range of architecture families across several domains, including image classification, neural machine translation (NMT) and language modeling, in addition to tasks from the BIG-Bench evaluation benchmark. Finally, we release a benchmark dataset comprising of 90 evaluation tasks to facilitate research in this domain.

## 1 Introduction

Scale has led to innovative research in both the vision domain [10, 14, 26, 35, 40] and the natural language processing (NLP) [8, 12] domain. Recent work has found that scaling up the data size [34], the model size [26, 35], the training schedule [5, 39] or all of them together [8, 40] often lead to improved performance. More importantly, scaling up the data size and the model size together can better utilize the compute resources. Scaling laws have been properly studied in several works, e.g. [3, 18–20, 23], and it has been found that the performance $f(x)$ (e.g. excess loss) often follows a power law $f(x) \sim \beta x^c$ for some $\beta > 0$ and $c < 0$ as one varies a dimension of interest $x$, such as the data or the model size.

While theoretical arguments alone seldom predict scaling law parameters in modern neural architectures [2, 21, 32], it has been observed that the benefit of scale could be predicted *empirically* [3, 4, 9, 17, 18, 20, 22, 23, 28, 30, 31]. The general approach is to acquire a *learning curve*, i.e. a collection of samples $(x, f(x))$, where $x$ is a dimension of interest such as the training data size while $f(x)$ is a measure of performance, such as the validation loss. After that, parameters are estimated, e.g. by computing the best-fitting values of $\beta$ and $c$ in the model $f(x) = \beta x^c$. Given the estimated scaling law parameters, one can then *extrapolate* by predicting $f(x)$ for large values of $x$.

Such learning curve extrapolation has found many applications, of which four seem to be more prominent. First, it offers a tool for understanding deep neural networks; e.g. how the architecture and data distribution impact scaling behaviors [1–3, 18, 20, 23, 30, 32]. Second, it has been used

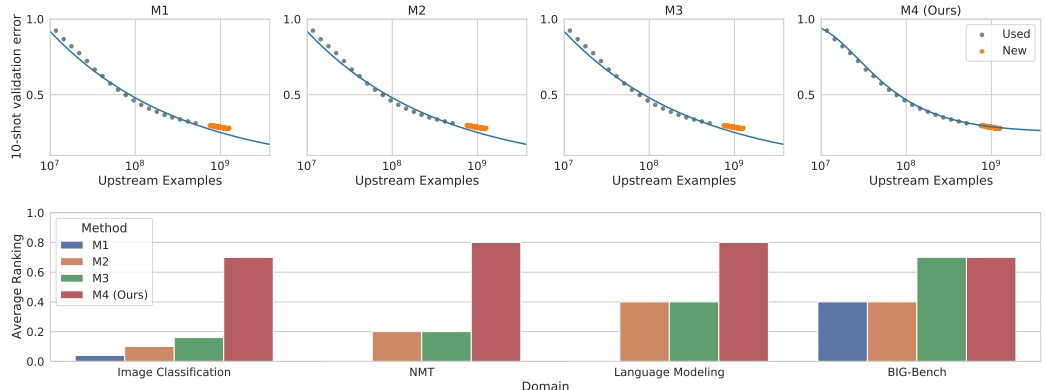

Figure 1: We introduce an estimator $\mathcal{M}_4$ (see Section 3) of scaling parameters that extrapolates more accurately from learning curves and compare it against previous methods denoted $\mathcal{M}_1$, $\mathcal{M}_2$, and $\mathcal{M}_3$ (see Section 2). **TOP**: The $y$-axis is ImageNet 10-shot error rate while the $x$-axis is the number of examples in JFT-300M [34] seen during pre-training. The architecture is BiT/101x3 [26] (see Section 5 for further details). Values in **amber** are not seen when fitting the scaling law. **BOTTOM**: Comparison across four domains. We report the fraction of time ($y$-axis, higher is better) in which a method achieves the best extrapolation error in the given domain's tasks (see Section 5). Because several methods may perform equally well in one task, average rankings do not always sum to one.

for sample size planning, particularly in data-scarce domains such as medicine [4, 9, 17, 28]. Third, it can reduce the environmental footprint of experimentation by terminating experiments early and accelerating hyper-parameter search [13, 20, 22]. Forth, it has been applied in neural architecture search (NAS) [15, 20, 25]. In addition, learning curve prediction offers a different methodology for comparing performance; e.g. instead of comparing accuracy on a single dataset, one can also examine the full (hypothetical) scaling curves.

However, in order to achieve such benefits in practice, it is imperative that scaling laws *extrapolate* accurately instead of merely interpolating the learning curve. To our knowledge, a validation of this sort based on extrapolation is often lacking in the literature and previous works have generally reported the best-fitting (interpolating) parameters. We demonstrate why this can be misleading in Section 4, where we illustrate how the scaling exponent $c$ that extrapolates best can be quite different from the exponent that best fits the given (finite) learning curve. In addition, we propose an estimator for scaling laws denoted $\mathcal{M}_4$, which extrapolates more accurately than previous methods as shown in Figure 1. We validate the proposed estimator in several domains, including image classification, neural machine translation, language modeling, and other related tasks.

**Our contributions are:**

1. We argue in Section 4 for a more rigorous methodology to validate scaling law parameters based on extrapolation, instead of only reporting the best-fitting (interpolating) parameters.

2. We propose a recipe in Section 3 to estimate scaling laws reliably from learning curves. The new estimator is verified across several domains: image classification, neural machine translation (NMT), language modeling, and tasks from BIG-Bench evaluation benchmark [6].

3. We use the proposed recipe to study the impact of the neural architecture's type and size on scaling exponents.

4. We release a benchmark dataset consisting of 90 tasks to accelerate research in scaling laws.

## 2 Related work

Power law scaling in deep neural architectures has been verified in a wide range of domains, including image classification [2, 20, 32, 40], language modeling [20, 23, 32], NMT [3, 18–20], and speech recognition [20]. To explain this theoretically, at least for data scaling, several works have argued

for a power law behavior under various contexts. For instance, in the universal learning setting [7] under the realizable case with a 0-1 misclassification loss, power law scaling emerges with exponent $c = 1$ if the hypothesis space has an infinite Littlestone tree but not an infinite VC-Littlestone tree [7]. Another argument for the exponent $c = 1$ can be made in the non-realizable setting if the chosen loss is sufficiently smooth and the model size is limited by deriving the variance of the empirical solution around its population limit [2]. A more relevant setting for deep neural networks is to assume that the model size is effectively infinite and the loss is Lipschitz continuous (e.g. continuous loss in bounded domains). Under the latter assumptions, it has been argued that the scaling exponent would satisfy $c = O(1/d)$ where $d$ is the intrinsic dimension of the data manifold [2, 21, 32]. This is consistent with the fact that scaling exponents often satisfy $c \ll 1$ and that the exponent seems to be independent of the neural network architecture size as long as the architecture is sufficiently large [20, 23, 32].

Writing $x$ for the dimension of interest (e.g. data size) and $\varepsilon_x$ for the error/loss of the model as a function of $x$, three function classes have been used in the literature to model the performance $\varepsilon_x$ as a function of $x$ while capturing its expected power law behavior:

$\mathcal{M}_1$ : The simplest model assumes a power law throughout the domain of $x$: $\varepsilon_x = \beta x^c$. This has been used, for example, to estimate the required sample size in healthcare [9], neural machine translation (NMT) [19], and language models [23], among others [20, 22, 32].

$\mathcal{M}_2$ : To capture saturating performance for large $x$ (i.e. when the Bayes optimal risk is bounded away from zero), a parameter $\varepsilon_\infty$ is added: $\varepsilon_x - \varepsilon_\infty = \beta x^c$. This is, perhaps, the most commonly used model in the literature; see for instance [1, 13, 17, 19, 20, 22, 28, 30, 31].

$\mathcal{M}_3$ : A different parameterization has been recently used in NMT [3]: $\varepsilon_x = \beta(x^{-1} + \gamma)^{-c}$, where $\beta > 0, \gamma \geq 0$ and $c < 0$. Variants of this approach were used previously in studying, for example, scaling laws in vision transformers [40], accelerating hyper-parameter optimization [13], and (more generally) in learning curve prediction [25].

In this work, we introduce a fourth estimator $\mathcal{M}_4$ and verify experimentally that it outperforms the above methods in terms of its extrapolation capability in several domains, as summarized in Figure 1. We describe $\mathcal{M}_4$ and discuss its rationale in Section 3.

## 3  The Scaling Law Estimator $\mathcal{M}_4$

**Motivation.**   The function class $\mathcal{M}_2$, in which it is assumed that $\varepsilon_x = \varepsilon_\infty + \beta x^c$, captures (by definition) what it means for the excess risk to follow a power law. Hence, a question naturally arises: do we need any other function classes to estimate the scaling law parameters $\varepsilon_\infty$, $\beta$ and $c$?

To see why using $\mathcal{M}_2$ can occasionally fail, consider the following simple classification problem whose optimal Bayes risk is known. Suppose that the instances $\mathbf{x} \in \mathbb{S}^{d-1}$ are generated uniformly at random from the unit sphere $\mathbb{S}^{d-1} = \{x \in \mathbb{R}^d : ||x||_2 = 1\}$. In addition, let the (ground-truth) labeling function be given by:

$$y(\mathbf{x}) = \begin{cases} \mathbf{sign}(\langle \mathbf{w}^\star, \mathbf{x} \rangle), & \text{with probability } 1 - \delta \\ -\mathbf{sign}(\langle \mathbf{w}^\star, \mathbf{x} \rangle), & \text{otherwise,} \end{cases}$$

for some fixed $\mathbf{w}^\star \in \mathbb{S}^{d-1}$. If a classifier is trained using, for example, logistic regression, the misclassification error rate $\varepsilon_x$ of the learning algorithm as a function of the data size $x$ would typically undergo three stages as illustrated in Figure 2 [20]. First, we have saturating performance for small sample sizes shown on the left, in which the trained model does not perform much better than random guessing. Second, we have a transitional stage in which the performance of the model improves quickly but it does not constitute a power law yet. Third, we have the final power law regime in which the excess risk $\varepsilon_x - \varepsilon_\infty$ fits a power law curve.

Let $\mathcal{D}_0 = \{(x, \varepsilon_x)\}$ be the learning curve and write $\mathcal{D}_\tau = \{(x, \varepsilon_x) : x \geq \tau\} \subseteq \mathcal{D}_0$ for the restriction of the learning curve to $x \geq \tau$. To extrapolate from a learning curve, we train each of the four models $\mathcal{M}_1, \mathcal{M}_2, \mathcal{M}_3$, and $\mathcal{M}_4$ on the learning curve after applying a cutoff $\tau$ to mitigate the effect of small data samples. Then, we plot the excess risk $\varepsilon_x - \varepsilon_\infty^\star$ predicted by each model, where $\varepsilon_\infty^\star = \delta$ is the (ground-truth) Bayes risk. Since $\varepsilon_\infty^\star$ is known exactly, an accurate model that extrapolates well would produce a *linear* curve in each plot. As shown in Figure 2, $\mathcal{M}_2$ is accurate only when the data resides entirely in the power law regime (rightmost figure), whereas $\mathcal{M}_4$ works well in all cases.

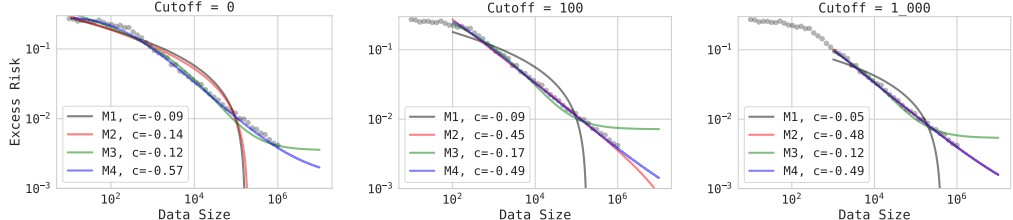

Figure 2: The excess risk $\varepsilon_x - \varepsilon_\infty^\star$ is plotted against the training data size for logistic regression where instances $\mathbf{x} \in \mathbb{R}^d$ are drawn uniformly at random from the surface of the unit sphere and the label is binary $\mathbf{y} \in \{-1, +1\}$ with noise rate $\delta$ (see Section 3). In this experiment, $d = 100$ and $\delta = 0.2$. In each figure, only the data sizes that exceed the indicated cutoff value are used to estimate the scaling law parameters. $\mathcal{M}_2$ is accurate only when the data resides entirely in the power law regime (rightmost figure), whereas $\mathcal{M}_4$ works well in all cases.

**Derivation.** The function class $\mathcal{M}_4$ arises from several natural requirements. First, we would like our function class to be sigmoid-like so that it fails only gracefully when the data deviates from the expected power law behavior; e.g. to avoid failures like that of $\mathcal{M}_1$ and $\mathcal{M}_2$ in Figure 2(left). Second, we would like our function class $\mathcal{F}$ to reduce to power law functions $f(x) \to \varepsilon_\infty + \beta x^c$ as $x \to \infty$. More precisely, we require that:

$$\forall f \in \mathcal{F} : \exists c < 0, \delta > 0 : \lim_{x \to \infty} \frac{\log(f(x) - \delta)}{\log x} = c. \tag{1}$$

To reiterate, this is because power law behavior has been empirically verified in a wide range of domains (see Section 2). Third, we would like our function class $\mathcal{F}$ to be expressive enough to contain all of the functions in $\mathcal{M}_2$, i.e. $\mathcal{M}_2 \subset \mathcal{M}_4$, so that using $\mathcal{M}_4$ becomes equivalent to using $\mathcal{M}_2$ when the observed learning curve resides entirely in the power law regime.

If we take the first requirement above on the shape of the function, a general approach to achieve this is to write the performance as a convex combination of the form:

$$\varepsilon_x = \gamma(x)(1 + \gamma(x))^{-1}\varepsilon_0 + (1 + \gamma(x))^{-1}\varepsilon_\infty, \tag{2}$$

for some function $\gamma(x)$ that satisfies $\lim_{x \to \infty} \gamma(x) = 0$ and $\lim_{x \to 0} \gamma(x) = \infty$. Here, $\varepsilon_\infty$ is the predicted limiting performance when $x \to \infty$ while $\varepsilon_0$ is the performance at the random-guessing level. To meet the second requirement, we set $\gamma(x) = \beta x^c$ for some learnable parameters $\beta > 0$ and $c < 0$. Rearranging the terms yields $(\varepsilon_x - \varepsilon_\infty) \cdot (\varepsilon_0 - \varepsilon_x)^{-1} = \gamma(x) \doteq \beta x^c$. Finally, we introduce a learnable parameter $\alpha > 0$ to meet our final requirement (see the ablation in Appendix A.1):

$$\frac{\varepsilon_x - \varepsilon_\infty}{(\varepsilon_0 - \varepsilon_x)^\alpha} = \beta x^c. \tag{$\mathcal{M}_4$}$$

With $\alpha = 0$, our function class $\mathcal{M}_4$ reduces to $\mathcal{M}_2$ as required. By differentiating both sides of the equation above and noting that $c < 0$, we deduce that $\varepsilon_x$ remains a monotone decreasing function of $x$ for all $\alpha \geq 0$ as expected. In addition, by rearranging terms and using both the binomial theorem and the Lagrange series inversion theorem, we have the following asymptotic expansion for the excess loss $\varepsilon_x - \varepsilon_\infty$ as $x \to \infty$:

$$\varepsilon_x - \varepsilon_\infty \sim (\varepsilon_0 - \varepsilon_\infty)^\alpha (\beta x^c) - \alpha(\varepsilon_0 - \varepsilon_\infty)^{2\alpha-1}(\beta x^c)^2 \tag{3}$$

When $\alpha = 0$, we recover the power law estimator $\mathcal{M}_2$. Setting $\alpha > 0$ allows $\mathcal{M}_4$ to handle measurements that deviate from the power law behavior; i.e. when the learning curve does not fall into the asymptotic power law regime. The difference is $O(\alpha x^{2c})$ (suppressing other constants).

The parameters to be fitted here are $\alpha \geq 0$, $\beta > 0$, $c < 0$ and $\varepsilon_\infty > 0$. The parameter $\varepsilon_0$ corresponds to the value of the loss at the random-guessing level and can be either fixed or optimized. We fix $\varepsilon_0$ in our evaluation to be equal to the loss at the random-guessing level, although we observe similar results when it is optimized.

**Loss Function.** In this work, scaling law parameters in all the four function classes are estimated by minimizing the square-log loss, similar to the approach used in [28]. This serves two purposes. First, it penalizes the *relative* loss and, hence, treats errors at all scales equally. Second, it allows us to compute a subset of the parameters in closed-form using least squares.

Specifically, in $\mathcal{M}_4$, for example, we minimize:

$$\mathcal{L}(\alpha, \beta, c, \varepsilon_\infty) = \mathbb{E}_{\mathbf{x}}\left[(\log(\varepsilon_x - \varepsilon_\infty) - \alpha\log(\varepsilon_0 - \varepsilon_x) - \log\beta - c\log x))^2\right]. \tag{4}$$

In $\mathcal{M}_2$, we optimize the same loss above while fixing $\alpha = 0$. In $\mathcal{M}_1$, we fix both $\alpha$ and $\varepsilon_\infty$ to zero. In $\mathcal{M}_3$, we optimize the following loss:

$$\mathcal{L}(\beta, c, \gamma) = \mathbb{E}_{\mathbf{x}}\left[\left(\log\varepsilon_x - \log\beta - c\log\left(x^{-1} + \gamma\right)\right)^2\right]. \tag{5}$$

In all function classes, we use block coordinate descent, where we compute $\alpha, \beta$ and $c$ in closed form using least squares, and estimate the remaining parameters (if any) using gradient descent, with a learning rate of $10^{-7}$. We repeat this until convergence.

## 4 Validating Scaling Laws using the Extrapolation Error

A common approach in the literature for estimating scaling law parameters is to assume a parametric model, e.g. $\mathcal{M}_2$, and reporting its best-fitting parameters to an empirical learning curve (see for example the prior works discussed in Section 2). Afterwards, patterns are reported about the behavior of the scaling law parameters; e.g. how the exponent varies with the architecture size. We argue, next, for a more rigorous methodology based on the *extrapolation* loss, instead of only reporting the best-fitting (interpolating) parameters. Specifically, choices of scaling law parameters that achieve a small interpolation error do not necessarily achieve a small extrapolation error so they may not, in fact, be valid estimates of scaling law parameters. Scaling law parameters should be validated by measuring how well they *extrapolate*.

To see why a validation of this sort matters, consider the following example. If we pretrain a vision transformer ViT/B/16 [14] on subsets of JFT-300M (a proprietary dataset with 300M examples and 18k classes [34]) using the Adam optimizer [24][1], and evaluate the 10-shot error rate on ImageNet-ILSRCV2012 [11], we obtain the learning curve shown in Figure 3(left, in green). Evidently, power law emerges; i.e. ImageNet 10-shot error rate (shown in green) follows a linear curve on a log-log plot. Hence, one might estimate, for example the scaling exponent $c$ using least squares.

However, consider now the family of curves shown in Figure 3(left), all corresponding to $\mathcal{M}_2$ but with scaling exponents that vary from about $c = -0.24$ to $c = -0.4$ (while fitting the parameters $\beta$ and $\varepsilon_\infty$). Note that all five curves overlap with each other significantly. Choosing the best fitting parameters on the learning curve would favor a scaling exponent of $c = -0.24$ as shown in Figure 3(right). However, if we validate the parameters by evaluating how well they *extrapolate* (i.e. how well they predict performance when the number of seen examples $x \gg 10^9$), a different picture emerges. We observe that a more accurate estimate of the scaling exponent is $c = -0.40$. This is shown in Figure 3(left) and in Figure 3(right) by measuring the extrapolation loss. Here, validation is measured using the root mean square error (RMSE) to the log-loss:

$$\mathrm{RMSE} = \sqrt{\mathbb{E}_{\mathbf{x}}(\log\hat{\varepsilon}_{\mathbf{x}} - \log\varepsilon_{\mathbf{x}})^2} \tag{6}$$

in which $\mathbf{x}$ is uniform over the set $[10^9, 2\times10^9]$, where $\hat{\varepsilon}_{\mathbf{x}}$ is the predicted loss while $\varepsilon_{\mathbf{x}}$ is the actual. We apply the logarithm so that we penalize the relative error and, hence, assign equal importance to all error scales (both large and small)[2].

In summary, scaling law parameters that give the best fit on the learning curve (i.e. lowest interpolation loss) do not generally extrapolate best. When using extrapolation loss instead, different scaling law parameters emerge. In this work, we use extrapolation to evaluate the quality of scaling law estimators.

---

[1]with a base learning rate of 5e-4, batch-size 4,096, and dropout rate of 0.1

[2]E.g. if we have a single measurement $x$ where $\varepsilon_x = 0.99$ and the estimator predicts $\hat{\varepsilon}_x = 1.0$, the RMSE in (6) is approximately equal to 1%. Similarly, it is approximately equal to 1% when $\varepsilon_x = 0.099$ while $\hat{\varepsilon}_x = 0.1$.

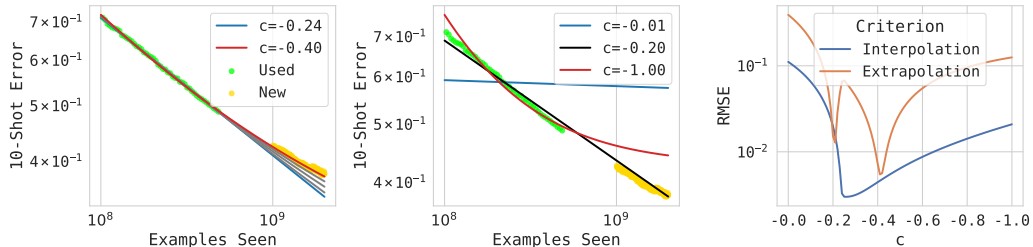

Figure 3: In this experiment, ViT/B/16 [14] is pretrained on JFT-300M [34], where the evaluation metric is 10-shot ImageNet error rate. **LEFT & MIDDLE**: the learning curve is plotted. Different scaling exponents using the function class $\mathcal{M}_2$ can fit the learning curve almost equally well. Values in green correspond to the data used to train the scaling law estimator (up to 500M seen examples) while values in yellow are used to evaluate the extrapolation loss. **RIGHT**: Best fitting parameters do not necessarily coincide with the scaling parameters that achieve small *extrapolation* loss.

Table 1: A list of the six architectures used in the evaluation study in the image classification domain.

| Residual Nets | | Vision Transformers | | MLP Mixers | |
|---|---|---|---|---|---|
| Model | # Parameters | Model | # Parameters | Model | # Parameters |
| BiT/50/1 | 61M | ViT/S/16 | 32M | MiX/B/16 | 73M |
| BiT/101/3 | 494M | ViT/B/16 | 110M | MiX/L/16 | 226M |

## 5 Experiments

We provide an empirical evaluation of the four scaling law estimators in several domains, including image classification (72 tasks), neural machine translation (5 tasks), language modeling (5 tasks), and other language-related evaluations (10 tasks). The dataset for neural machine translation is available at [3]. The code and dataset for the remaining tasks used in this evaluation are made publicly available to facilitate further research in this domain[3]. In all experiments, we divide the learning curve into two splits: (1) one split used for training the scaling law estimators, and (2) one split used for evaluating extrapolation. Setting $\tau = x_{max}/2$, where $x_{max}$ is the maximum value of $x$ in the data, the first split is the domain $x \in [0, \tau]$ while the second split is the domain $x \in (\tau, 2\tau]$. We measure extrapolation error using RMSE in (6). All experiments are executed on Tensor Processing Units (TPUs).

### 5.1 Image Classification

**Architectures and Tasks.** We use three families of architectures: (1) big-transfer residual neural networks (BiT) [26], (2) vision transformers (ViT) [14], and (3) MLP mixers (MiX) [37]. For each family, we have two models of different sizes as shown in Table 1 in order to assess the impact of the size of the architecture on the scaling parameters. We pretrain on JFT-300M [34]. Since pre-training task performance is not representative of the downstream performance [40], we evaluate the few-shot accuracy downstream on four datasets: (1) ImageNet [11], (2) Birds 200 [38], (3) CIFAR100 [27], and (4) Caltech101 [16]. For each dataset, we report 5/10/25-shot accuracy. It results in 72 tasks for the combinations of architecture, dataset, and metric. Following [14, 26], we removed duplicate pre-training examples between upstream JFT-300M dataset and all the downstream train and test sets.

**Bootstrapped Examples.** In the few-shot image classification setting under the transfer learning setup, overfitting can occur if the upstream dataset is small, where training beyond a particular number of steps would reduce the downstream validation accuracy . This is demonstrated in Figure 4, where we pretrain on subsets of JFT-300M (upstream) and evaluate ImageNet 10-shot error (downstream).

Nevertheless, we observe that prior to reaching peak performance, training examples behave as if they were fresh samples. This observation generalizes the bootstrapping phenomenon observed in

---

[3]Code and benchmark dataset will be made available at: `https://github.com/google-research/google-research/tree/master/revisiting_neural_scaling_laws`.

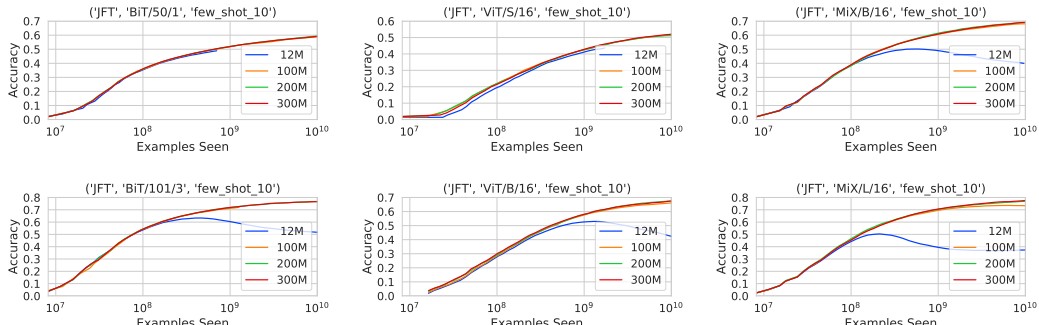

Figure 4: In this experiment, we pretrain each vision architecture on subsets of JFT-300M [34] and report the 10-shot ImageNet accuracy [11]. When the architecture is pretrained on a subset of size 12M, we observe overfitting, where the performance begins to *drop* if the model is pretrained for a large number of steps. Nevertheless, prior to reaching peak performance, training examples behave as if they are fresh samples, which is consistent with the earlier observations reported in [29].

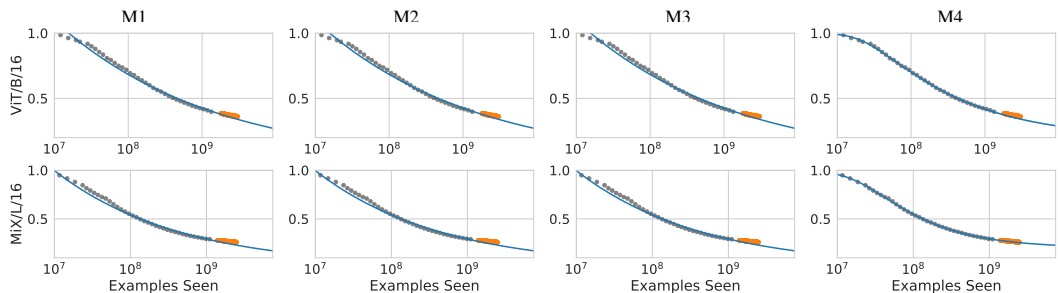

Figure 5: ImageNet 10-shot accuracy ($y$-axis) vs. the number of bootstrapped examples seen during upstream training in ViT/B/16 and MiX/L/16 (see Figure 1 for BiT/101x3 and Appendix A.3 for all remaining figures). The curves in each column correspond to the scaling law learned using the corresponding function class. The values marked in amber are reserved for evaluating how well the scaling law parameters extrapolate. Generally, $\mathcal{M}_4$ extrapolates better than previous methods.

[29], where it showed that training examples behave as fresh samples prior to convergence, which would be equivalent to our observation if no overfitting occurs. Throughout the sequel, we refer to the examples seen during training prior to peak performance as "bootstrapped examples" and use their number as the independent variable $x$ when evaluating the scaling law estimators in this section.

Figure 5 illustrates how well each of the the four scaling law estimators $\mathcal{M}_1$, $\mathcal{M}_2$, $\mathcal{M}_3$, and $\mathcal{M}_4$ can extrapolate from a given learning curve. The complete set of figures is provided in Appendix A.3. We observe that $\mathcal{M}_4$ extrapolates better than other methods and produces learning curves that approximate the empirical results more faithfully. As shown in Figure 1, $\mathcal{M}_4$ outperforms the other methods in more than 70% of the tasks in this domain.

**Impact of the Architecture.** Figure 6 plots the scaling exponent $c$ in each architecture when the downstream task is $n$-shot accuracy on ImageNet. We observe that within each family of models, larger models have more favorable scaling exponents. In addition, $\mathcal{M}_4$ yields estimates of the scaling exponents that are larger in absolute magnitude than in other methods. Figure 7 shows that such differences in scaling exponents show up indeed in the slopes of the learning curves as expected.

### 5.2 Neural Machine Translation (NMT)

Next, we evaluate the scaling law estimators on NMT. We use the setup studied in [3], in which models are trained with the per-token cross-entropy loss using Adafactor optimizer [33] with a batch-size of 500K tokens and a dropout rate of 0.1 [3]. We use the encoder-decoder transformer models 6L6L, 28L6L and 6L28L, where 28L6L means that the architecture has 28 encoders and 6 decoders. We also use the two architectures: decoder-only with language modeling loss (D/LM) and the transformer-encoder with LSTM decoder (TE/LSTM). These correspond to the architectures used

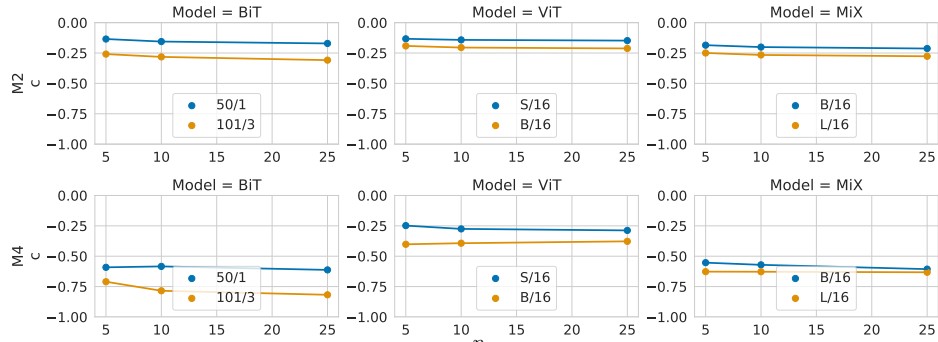

Figure 6: The scaling exponent $c$ is plotted for each architecture when pretraining on subsets of JFT-300M and evaluating performance using $n$-shot accuracy on ImageNet, where $n$ is the $x$-axis. Top row is for $\mathcal{M}_2$ (nearly identical in this case to $\mathcal{M}_3$) while the bottom row is for $\mathcal{M}_4$. We observe that $\mathcal{M}_4$ suggests more favorable scaling behavior (larger estimates of the scaling exponents) than in previous methods. Also, larger architectures within the same family have more favorable exponents.

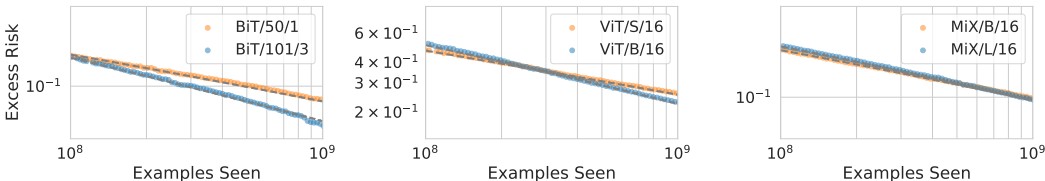

Figure 7: The excess risk of 10-shot ImageNet classification when pretraining on JFT-300M is plotted against the number of examples seen upstream. The slope of each curve is its scaling exponents $c$.

in Figure 1 in [3]. In all cases, performance is measured using log-perplexity on a hold-out dataset. To evaluate the accuracy of the scaling law estimator, we fit its parameters on the given learning curve (for up to 256M sentence pairs) and use it to predict the log-perplexity when the architecture is trained on 512M sentence pairs. Because the learning curves contain few points, we only evaluate on the 512M sentence pairs. Table 2 displays the RMSE of each estimator. Clearly, $\mathcal{M}_4$ performs better than the other methods as summarized in Figure 1, which is consistent with the earlier results in image classification.

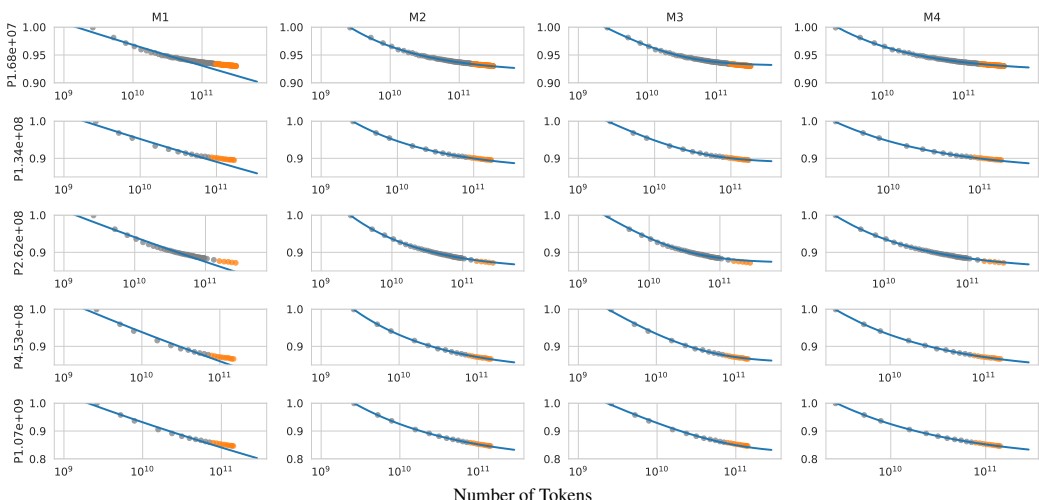

Figure 8: We evaluate scaling law estimators on language modeling tasks (see Section 5.3) with various model sizes indicated in the left side of each row. Table 2 provides the RMSE scores.

Table 2: Extrapolation RMSE of the scaling law estimators on the five NMT tasks (top) and the five language modeling tasks. See Sections 5.2 and 5.3 for further details.

| Model | M1 | M2 | M3 | M4 |
|---|---|---|---|---|
| | | NMT | | |
| 6 Enc, 6 Dec Layers | $2.6 \times 10^{-1}$ | $3.9 \times 10^{-2}$ | $8.9 \times 10^{-2}$ | $\mathbf{1.0 \times 10^{-2}}$ |
| 28 Enc, 6 Dec Layers | $1.7 \times 10^{-1}$ | $5.6 \times 10^{-2}$ | $3.3 \times 10^{-2}$ | $\mathbf{1.3 \times 10^{-2}}$ |
| 6 Enc, 28 Dec Layers | $2.3 \times 10^{-1}$ | $5.3 \times 10^{-2}$ | $\mathbf{1.6 \times 10^{-2}}$ | $3.0 \times 10^{-2}$ |
| Decoder-only /LM | $2.5 \times 10^{-1}$ | $3.9 \times 10^{-2}$ | $8.9 \times 10^{-2}$ | $\mathbf{1.0 \times 10^{-2}}$ |
| Transformer Enc /LSTM Dec | $1.9 \times 10^{-1}$ | $\mathbf{1.3 \times 10^{-2}}$ | $6.2 \times 10^{-2}$ | $1.2 \times 10^{-2}$ |
| | | Language Modeling | | |
| 1.68e+07 | $1.5 \pm 0.1 \times 10^{-2}$ | $6.0 \pm 1.0 \times 10^{-4}$ | $2.5 \pm 0.2 \times 10^{-3}$ | $\mathbf{3.1 \pm 0.7 \times 10^{-4}}$ |
| 1.34e+08 | $1.6 \pm 0.3 \times 10^{-2}$ | $1.7 \pm 0.4 \times 10^{-3}$ | $\mathbf{6.6 \pm 3.0 \times 10^{-4}}$ | $1.9 \pm 0.4 \times 10^{-3}$ |
| 2.62e+08 | $2.3 \pm 0.5 \times 10^{-2}$ | $\mathbf{1.9 \pm 0.5 \times 10^{-3}}$ | $5.2 \pm 0.9 \times 10^{-3}$ | $\mathbf{1.8 \pm 0.5 \times 10^{-3}}$ |
| 4.53e+08 | $1.7 \pm 0.4 \times 10^{-2}$ | $7.4 \pm 5.5 \times 10^{-4}$ | $6.6 \pm 3.8 \times 10^{-4}$ | $\mathbf{7.5 \pm 5.7 \times 10^{-4}}$ |
| 1.07e+09 | $1.7 \pm 0.4 \times 10^{-2}$ | $1.7 \pm 0.3 \times 10^{-3}$ | $4.5 \pm 0.4 \times 10^{-3}$ | $\mathbf{1.3 \pm 0.2 \times 10^{-3}}$ |

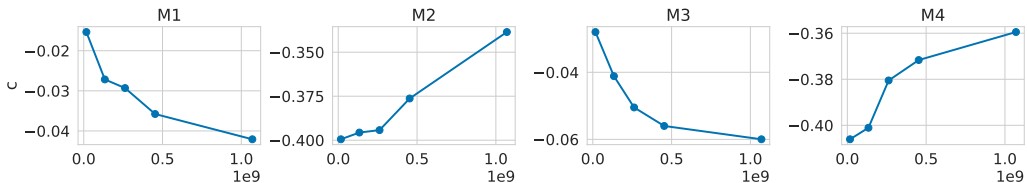

Figure 9: Scaling exponent $c$ is plotted against the language model size. Unlike in image classification with transfer learning (see Figure 6), $|c|$ seems to decrease using $\mathcal{M}_4$ as the model size increases.

Table 3: A summary of BIG-Bench evaluation results using the extrapolation RMSE in (6). See Section 5.4 for further details. Both $\mathcal{M}_3$ and $\mathcal{M}_4$ perform best in this domain.

| | M1 | M2 | M3 | M4 |
|---|---|---|---|---|
| linguistic_mappings: 1-shot | $\mathbf{1.6 \pm 0.2 \times 10^{-2}}$ | $\mathbf{1.6 \pm 0.1 \times 10^{-2}}$ | $\mathbf{1.6 \pm 0.2 \times 10^{-2}}$ | $1.7 \pm 0.2 \times 10^{-2}$ |
| linguistic_mappings: 2-shot | $1.7 \pm 0.2 \times 10^{-2}$ | $1.7 \pm 0.2 \times 10^{-2}$ | $1.7 \pm 0.2 \times 10^{-2}$ | $\mathbf{9.2 \pm 1.4 \times 10^{-3}}$ |
| qa_wikidata: 1-shot | $\mathbf{4.2 \pm 2.3 \times 10^{-3}}$ | $4.4 \pm 2.1 \times 10^{-3}$ | $\mathbf{4.2 \pm 2.3 \times 10^{-3}}$ | $4.4 \pm 2.1 \times 10^{-3}$ |
| qa_wikidata: 2-shot | $\mathbf{4.4 \pm 1.9 \times 10^{-3}}$ | $4.7 \pm 1.7 \times 10^{-3}$ | $\mathbf{4.4 \pm 1.9 \times 10^{-3}}$ | $4.9 \pm 1.7 \times 10^{-3}$ |
| unit_conversion: 1-shot | $8.3 \pm 1.3 \times 10^{-3}$ | $8.1 \pm 1.3 \times 10^{-3}$ | $\mathbf{1.5 \pm 0.7 \times 10^{-3}}$ | $2.3 \pm 0.6 \times 10^{-3}$ |
| unit_conversion: 2-shot | $1.1 \pm 0.1 \times 10^{-2}$ | $1.1 \pm 0.1 \times 10^{-2}$ | $\mathbf{7.5 \pm 0.2 \times 10^{-3}}$ | $2.9 \pm 1.2 \times 10^{-3}$ |
| mult_data_wrangling: 1-shot | $\mathbf{1.1 \pm 0.3 \times 10^{-2}}$ | $\mathbf{1.1 \pm 0.3 \times 10^{-2}}$ | $\mathbf{1.1 \pm 0.3 \times 10^{-2}}$ | $1.3 \pm 0.3 \times 10^{-2}$ |
| mult_data_wrangling: 2-shot | $1.6 \pm 0.4 \times 10^{-2}$ | $1.6 \pm 0.4 \times 10^{-2}$ | $1.6 \pm 0.4 \times 10^{-2}$ | $\mathbf{6.2 \pm 2.2 \times 10^{-3}}$ |
| date_understanding: 1-shot | $3.2 \pm 0.3 \times 10^{-2}$ | $3.2 \pm 0.3 \times 10^{-2}$ | $\mathbf{4.7 \pm 0.4 \times 10^{-3}}$ | $1.5 \pm 1.3 \times 10^{-2}$ |
| date_understanding: 2-shot | $2.9 \pm 1.9 \times 10^{-2}$ | $2.9 \pm 1.9 \times 10^{-2}$ | $\mathbf{4.8 \pm 1.2 \times 10^{-3}}$ | $1.8 \pm 1.6 \times 10^{-2}$ |

## 5.3 Language Modeling

Next, we evaluate the scaling law estimators in language modeling, where the goal is to predict the next token. We use the LaMDA architecture used in [36], which is a decoder-only transformer language model. Five model sizes are used, ranging from $10^7$ to $10^9$ model parameters. In each model, we rescale validation loss to the unit interval $[0, 1]$. Figure 8 and Table 2 summarize the results. We observe that $\mathcal{M}_4$ and $\mathcal{M}_2$ perform best, with $\mathcal{M}_4$ tending to perform better. As stated earlier, $\mathcal{M}_4$ becomes equivalent to $\mathcal{M}_2$ when the learning curve resides entirely in the power law regime, hence the similar performance. Figure 9 displays the scaling exponents $c$ predicted by each estimator as a function of the architecture size. We observe that $\mathcal{M}_1$ and $\mathcal{M}_3$ produce estimates of $c$ that are small in absolute magnitude. However, in $\mathcal{M}_2$ and $\mathcal{M}_4$, the scaling exponent is close to $-1/3$ and decreases (in absolute magnitude) for larger models.

### 5.4 Scalable Tasks from the BIG-Bench Evaluation Benchmark

Finally, we evaluate the scaling law estimators on language tasks from the BIG-Bench collaborative benchmark [6]. Here, we pretrain a 262M-parameter decoder-only transformer (middle architecture in Figure 8) on language modeling and evaluate its 1-shot and 2-shot capabilities in five language-related tasks. We choose the five tasks that exhibit the highest learnability from the benchmark (i.e. improvement in performance when pretrained on language modeling, see [6] for details). The five tasks are: `linguistic_mappings`, `qa_wikidata`, `unit_conversion`, `mult_data_wrangling` and `date_understanding`. We use the benchmark's preferred metrics in all cases, which is either "multiple choice grade" or "exact string match" depending on the task. Table 3 and Figure 1 summarize the results. In this evaluation, both $\mathcal{M}_3$ and $\mathcal{M}_4$ perform best and equally well. In addition, we observe that $\mathcal{M}_1$ and $\mathcal{M}_2$ perform equally well and consistently worse than the other methods. One possible reason is that the learning curves are quite noisy (see Appendix A.2).

## 6 Discussion

The remarkable progress in deep learning in recent years is largely driven by improvements in scale, where bigger models are trained on larger datasets for longer training schedules. Several works observe that the benefit of scale can be predicted empirically by extrapolating from learning curves and this has found important applications, such as in sample size planning and neural architecture search. However, to achieve such benefits in practice, it is imperative that scaling laws extrapolate accurately. We demonstrate that scaling parameters that yield the best fit to the learning curve do not generally extrapolate best, thereby challenging their use as valid estimate of scaling law parameters. Hence, we argue for a more rigorous validation of scaling law parameters based on the extrapolation loss. In addition, we present a recipe for estimating scaling law parameters that extrapolates more accurately than in previous works, which we verify in several state-of-the-art architecture across a wide range of domains. To facilitate research in this domain, we also release a benchmark dataset comprising of 90 evaluation tasks. We believe that the proposed scaling law estimator can be utilized, for example, to accelerate neural architecture search (NAS), which we plan to study in future work.

## Acknowledgements

The authors would like to acknowledge and thank Behrooz Ghorbani for his feedback on earlier drafts of this manuscript as well as Ambrose Slone and Lucas Beyer for their help with some of the experiments. We also would like to thank Daniel Keysers and Olivier Bousquet for the useful discussions.

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
