BiT/101/3, MiX/L/16 and ViT/B/16 (see Figures 1 and 5 in the main paper). Here, we evaluate a version of the $\mathcal{M}_4$ estimator that does not contain the $\alpha$ parameter for a comparison. See Section A.1.

## A  Appendix

### A.1  Ablation

In this section, we show that the improvement in $\mathcal{M}_4$ compared to $\mathcal{M}_2$ is due to both of the conditions discussed in Section 3: (1) $\mathcal{M}_4$ has a sigmoid-like shape so that it can handle deviations from the power law behavior and (2) it contains all of the functions in $\mathcal{M}_2$ so that $\mathcal{M}_4$ reduces to $\mathcal{M}_2$ when the learning curve falls entirely in the power law regime.

First, we observe that the second condition alone is not sufficient to have a high extrapolation accuracy since $\mathcal{M}_2$ itself has a less extrapolation accuracy than $\mathcal{M}_4$. To show that a sigmoid-like behavior is not sufficient, we evaluation a different version of $\mathcal{M}_4$, in which $\alpha$ is not introduced. Recall that the parameter $\alpha$ was introduced so that $\mathcal{M}_4$ contains $\mathcal{M}_2$. Figure 10 and Table 4 present the extrapolation performance of all the estimators when using 10-shot ImageNet accuracy as a metric. We observe that $\mathcal{M}_4$ without the $\alpha$ parameter does not perform as well as when $\alpha$ is introduced.

Table 4: Extrapolation RMSE of the scaling law estimators in image classification using 10-shot ImageNet metric for $\mathcal{M}_4$ without the $\alpha$ parameter compared to the proposed $\mathcal{M}_4$ estimator.

|  | without $\alpha$ | M4 (ours) |
| --- | --- | --- |
| BiT/101/3 | $5.1 \pm 0.10 \times 10^{-2}$ | $\mathbf{1.7 \pm 0.05 \times 10^{-2}}$ |
| BiT/50/1 | $3.1 \pm 0.04 \times 10^{-2}$ | $\mathbf{0.9 \pm 0.02 \times 10^{-2}}$ |
| MiX/B/16 | $4.6 \pm 0.05 \times 10^{-2}$ | $\mathbf{1.3 \pm 0.02 \times 10^{-2}}$ |
| MiX/L/16 | $4.4 \pm 0.06 \times 10^{-2}$ | $\mathbf{1.0 \pm 0.03 \times 10^{-2}}$ |
| ViT/B/16 | $4.3 \pm 0.06 \times 10^{-2}$ | $\mathbf{2.2 \pm 0.04 \times 10^{-2}}$ |
| ViT/S/16 | $5.1 \pm 0.04 \times 10^{-2}$ | $\mathbf{3.1 \pm 0.04 \times 10^{-2}}$ |

### A.2  Big Bench Learning Curves

In Figure 11, we plot the learning curves in the BIG-bench evaluation tasks. We observe that the learning curves are noisier in this setting than in previous cases.

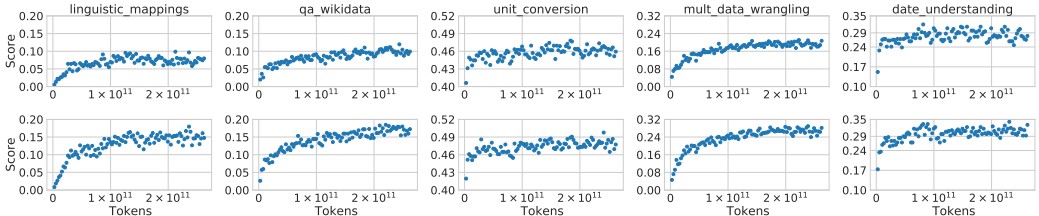

Figure 11: Learning curves are plotted for five tasks in BIG-Bench [6] when pretraining a 262M decoder-only architecture on language modeling. The benchmark's preferred score is used: "multiple choice grade" for `unit_conversion` and `date_understanding`, and "string match", otherwise.

## A.3  Image Classification Full Figures

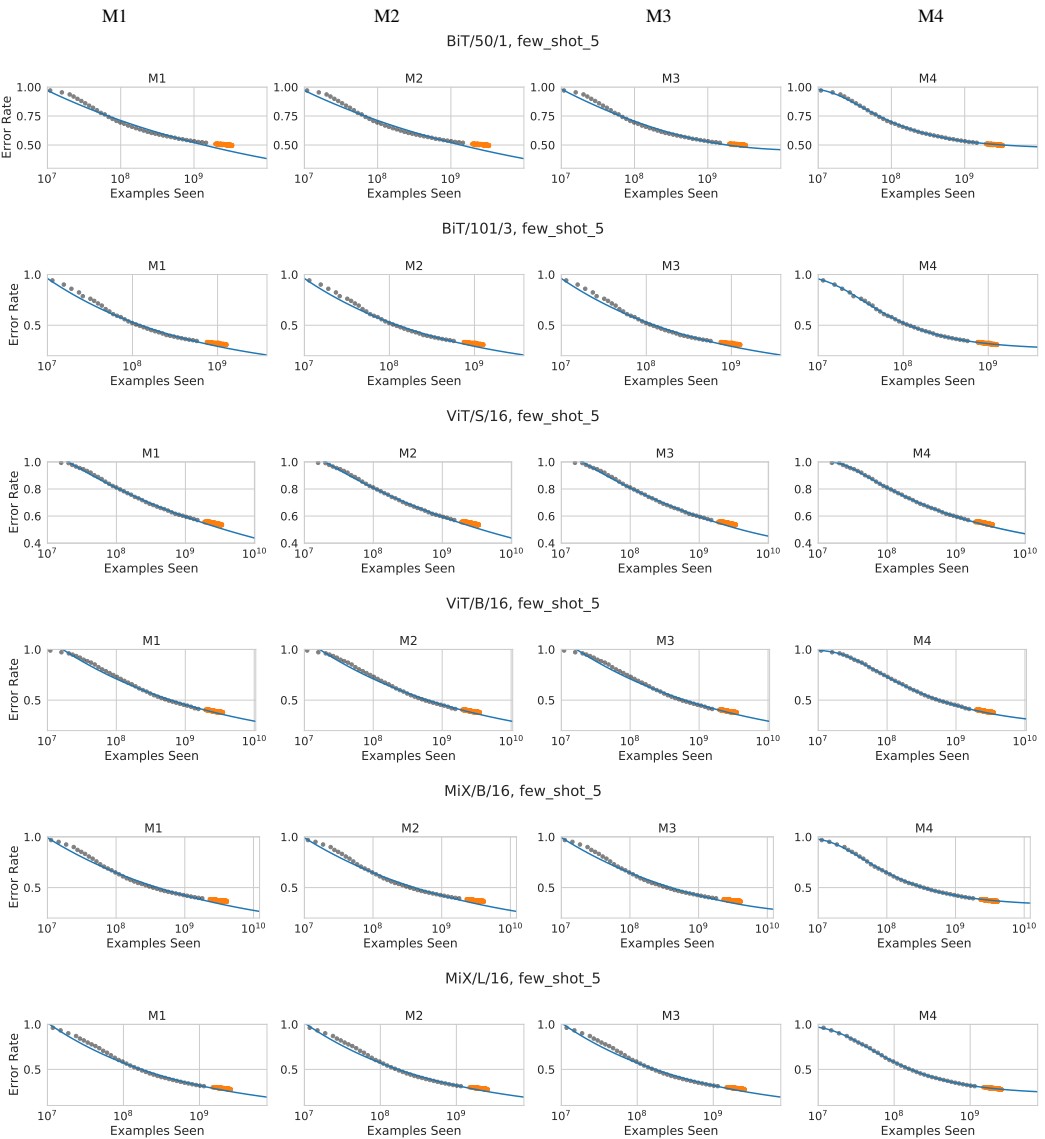

Figure 12: ImageNet 5-shot.

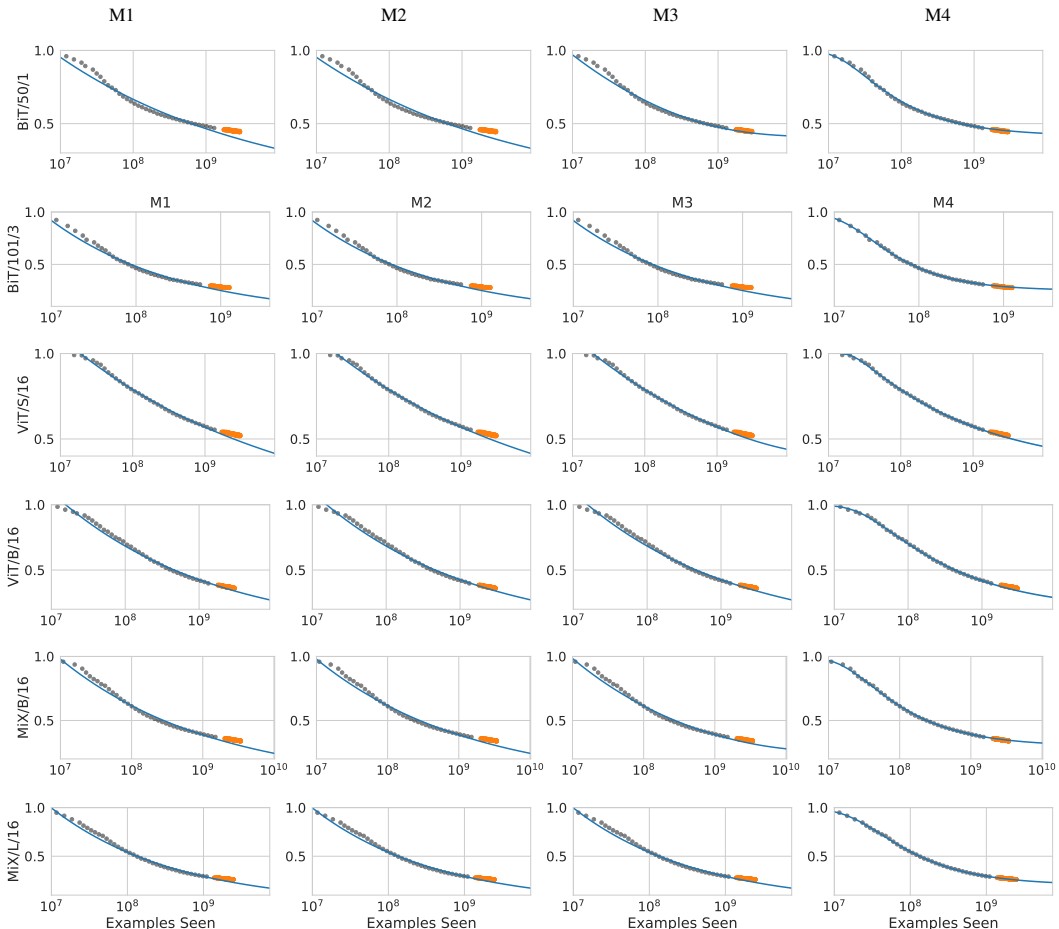

Figure 13: ImageNet 10-shot.

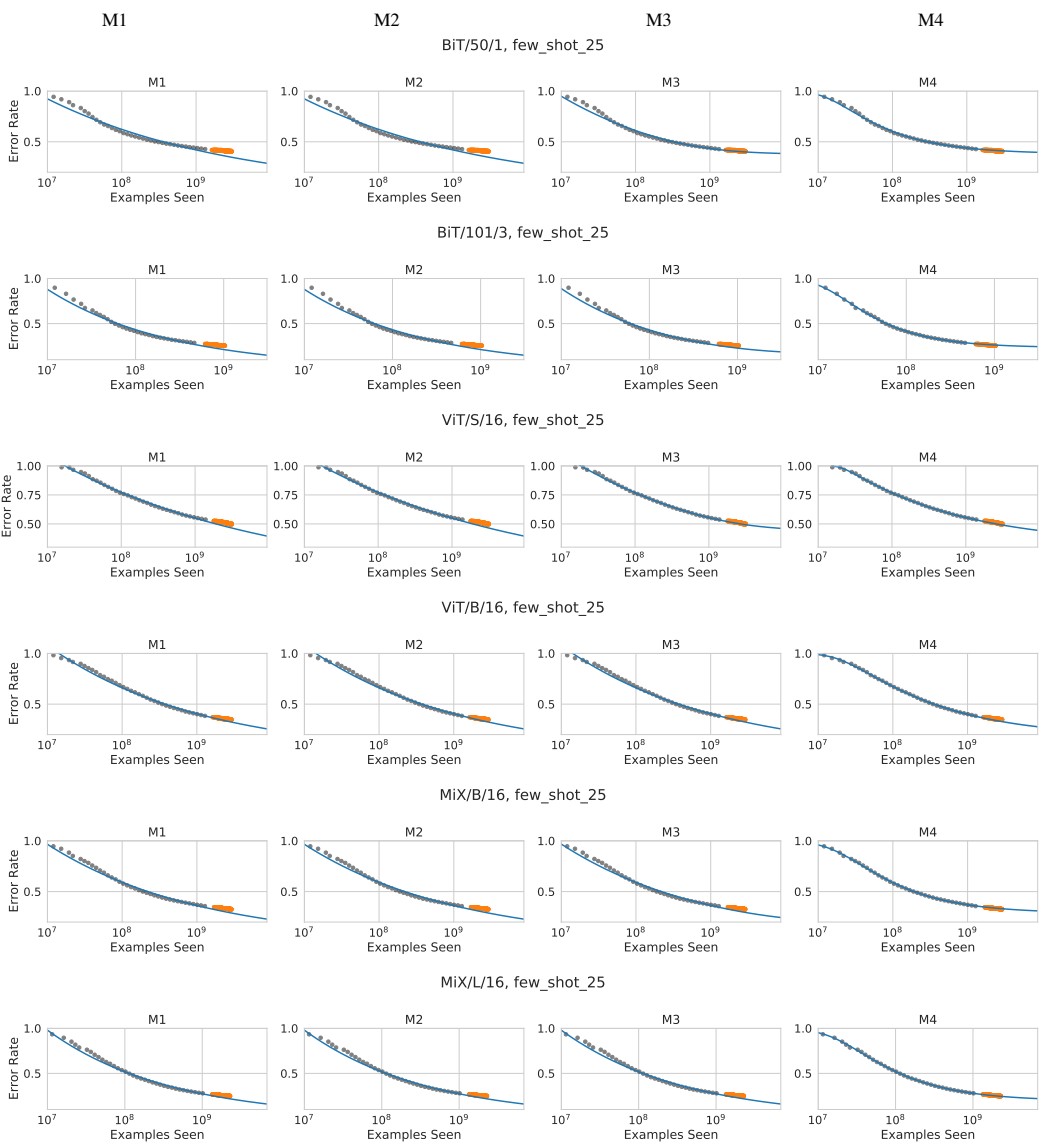

Figure 14: ImageNet 25-shot.

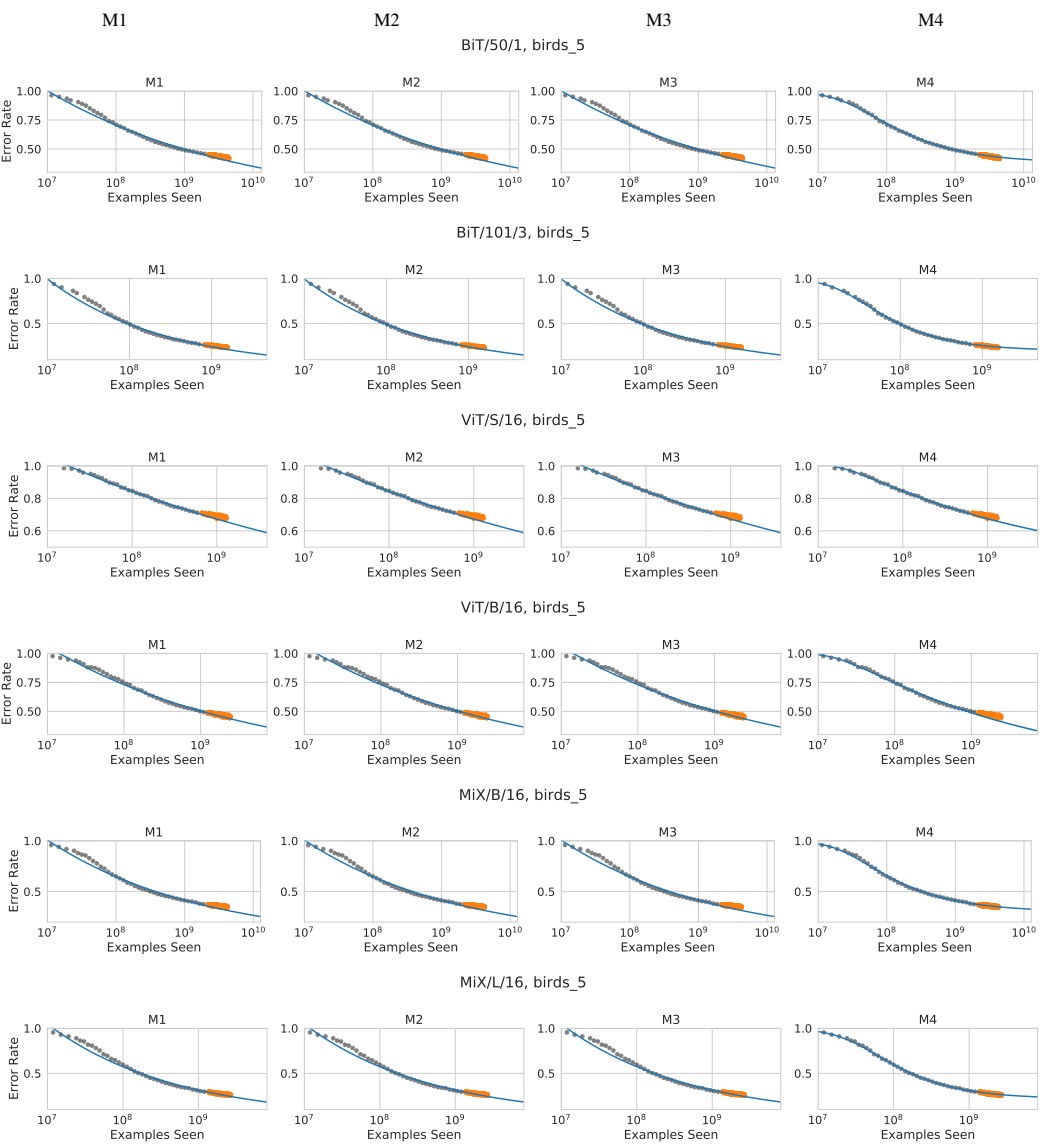

Figure 15: Birds2010 5-shot.

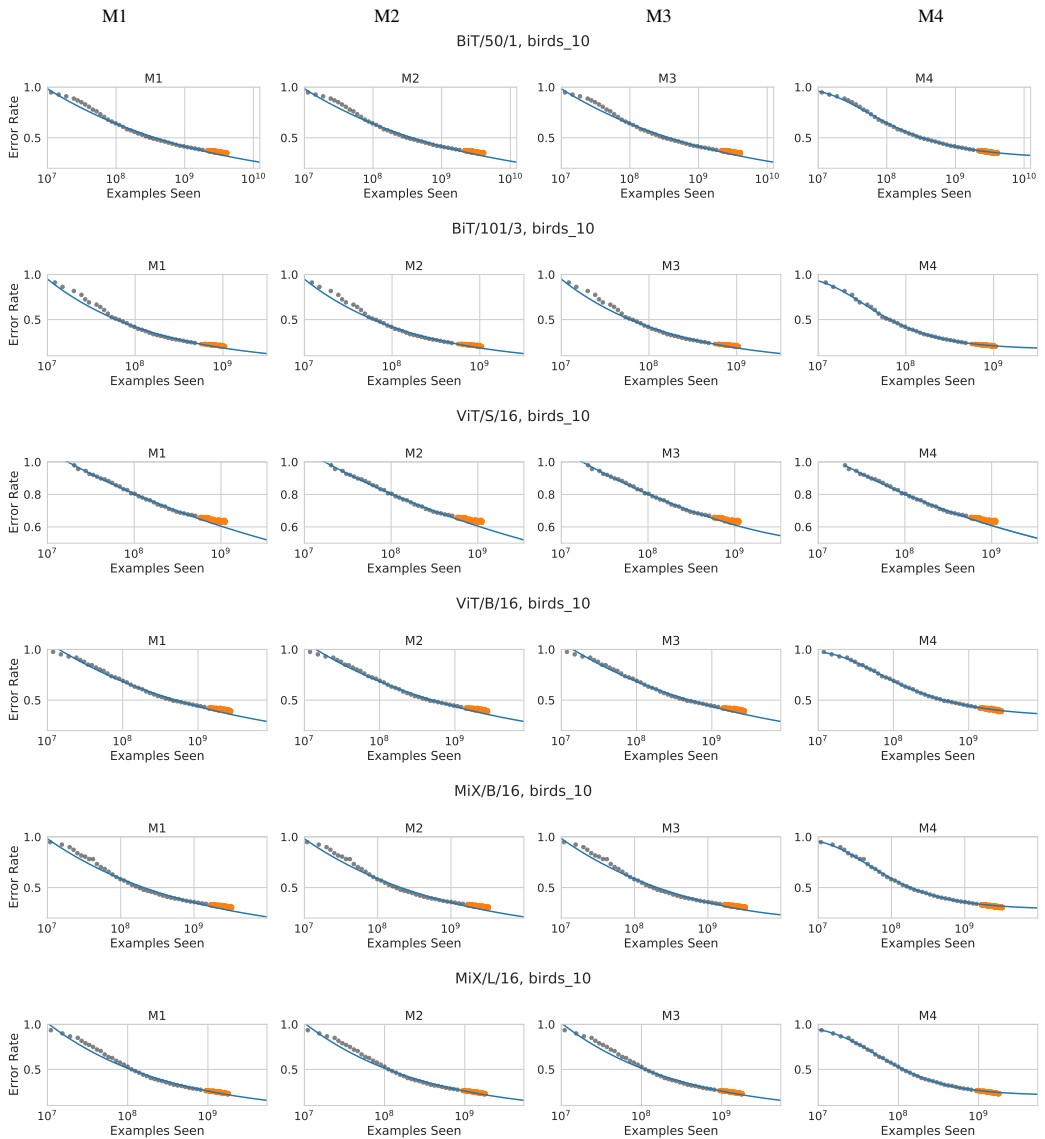

Figure 16: Birds2010 10-shot.

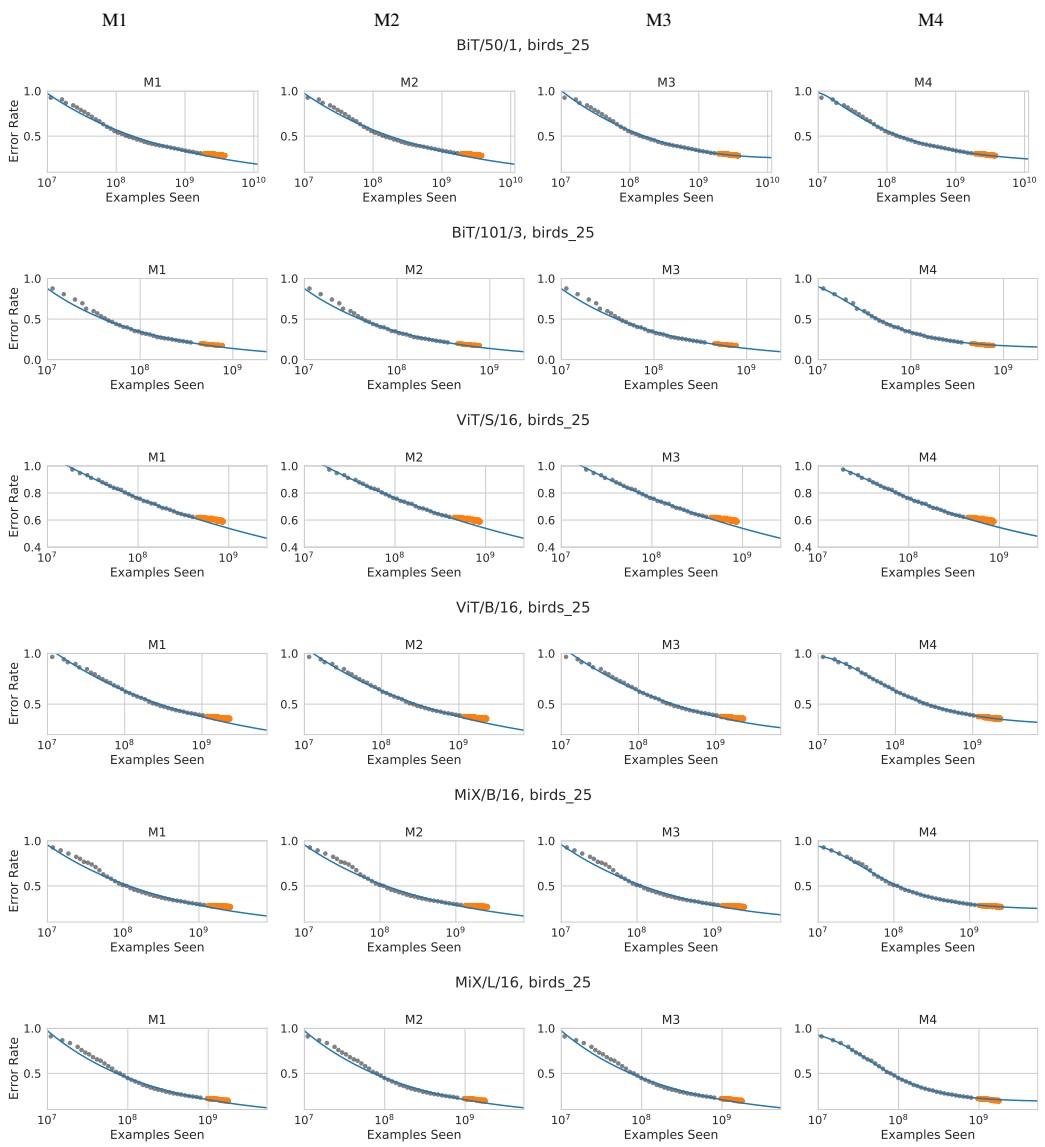

Figure 17: Birds2010 25-shot.

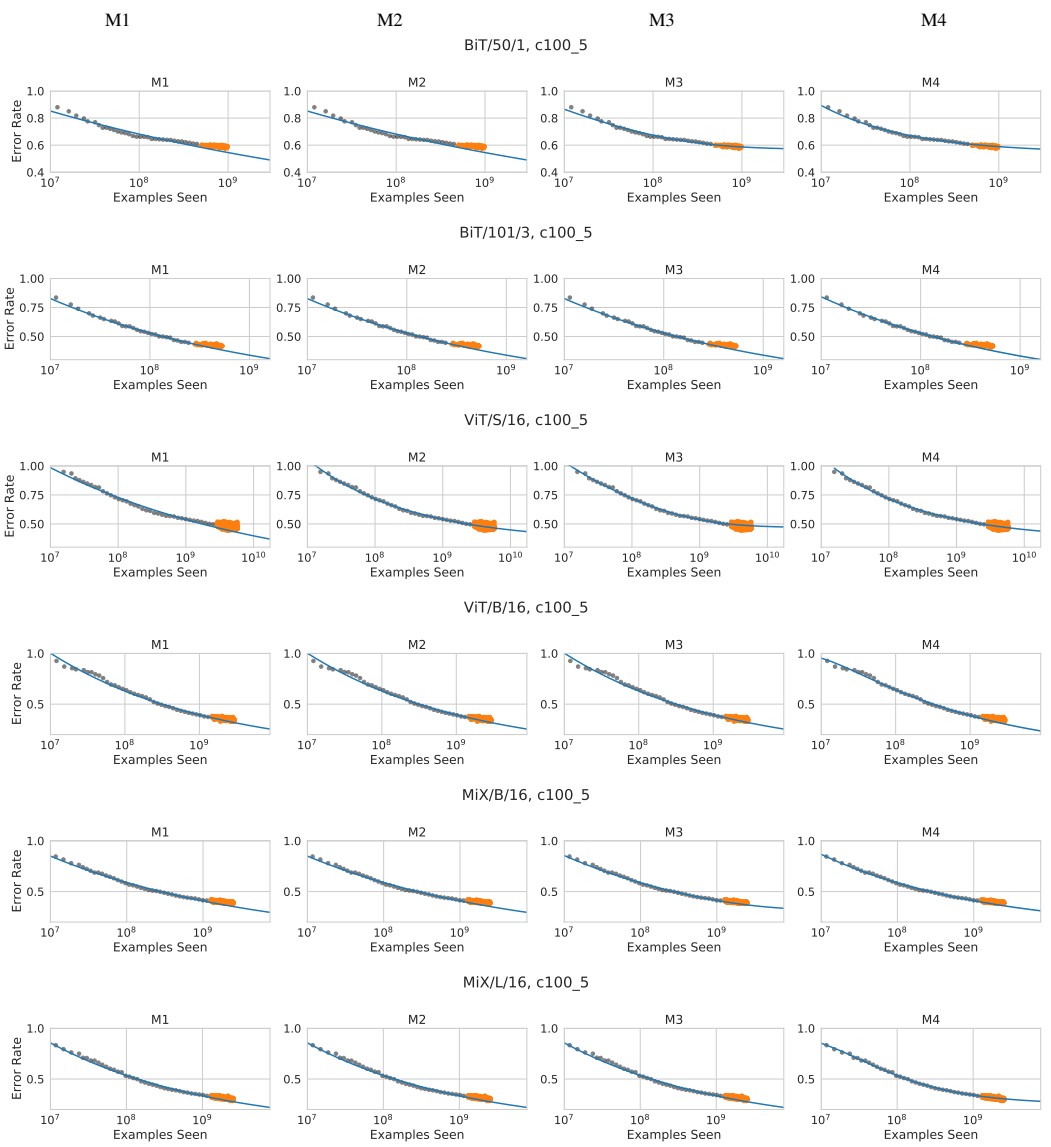

Figure 18: CIFAR100 5-shot.

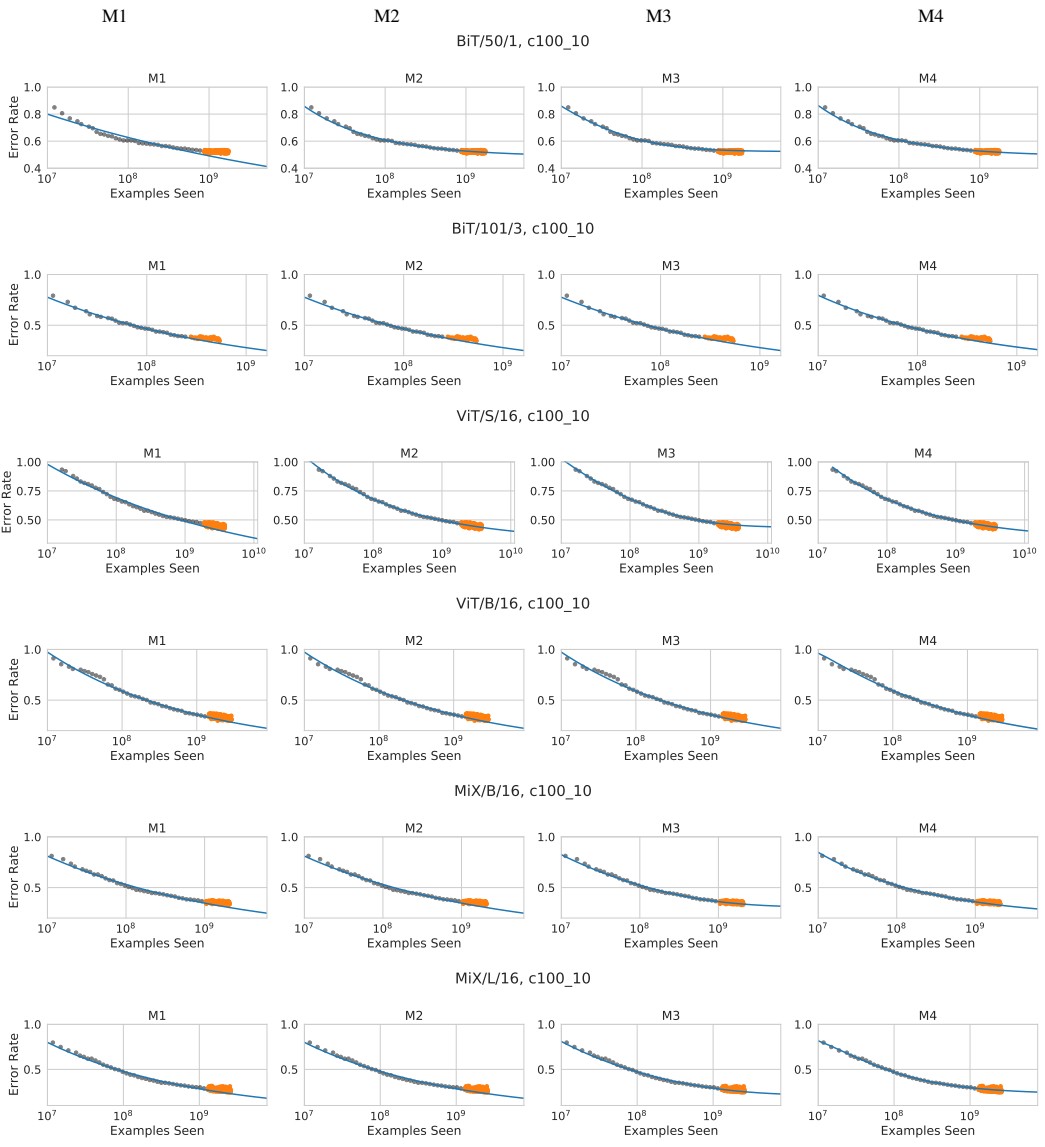

Figure 19: CIFAR100 10-shot.

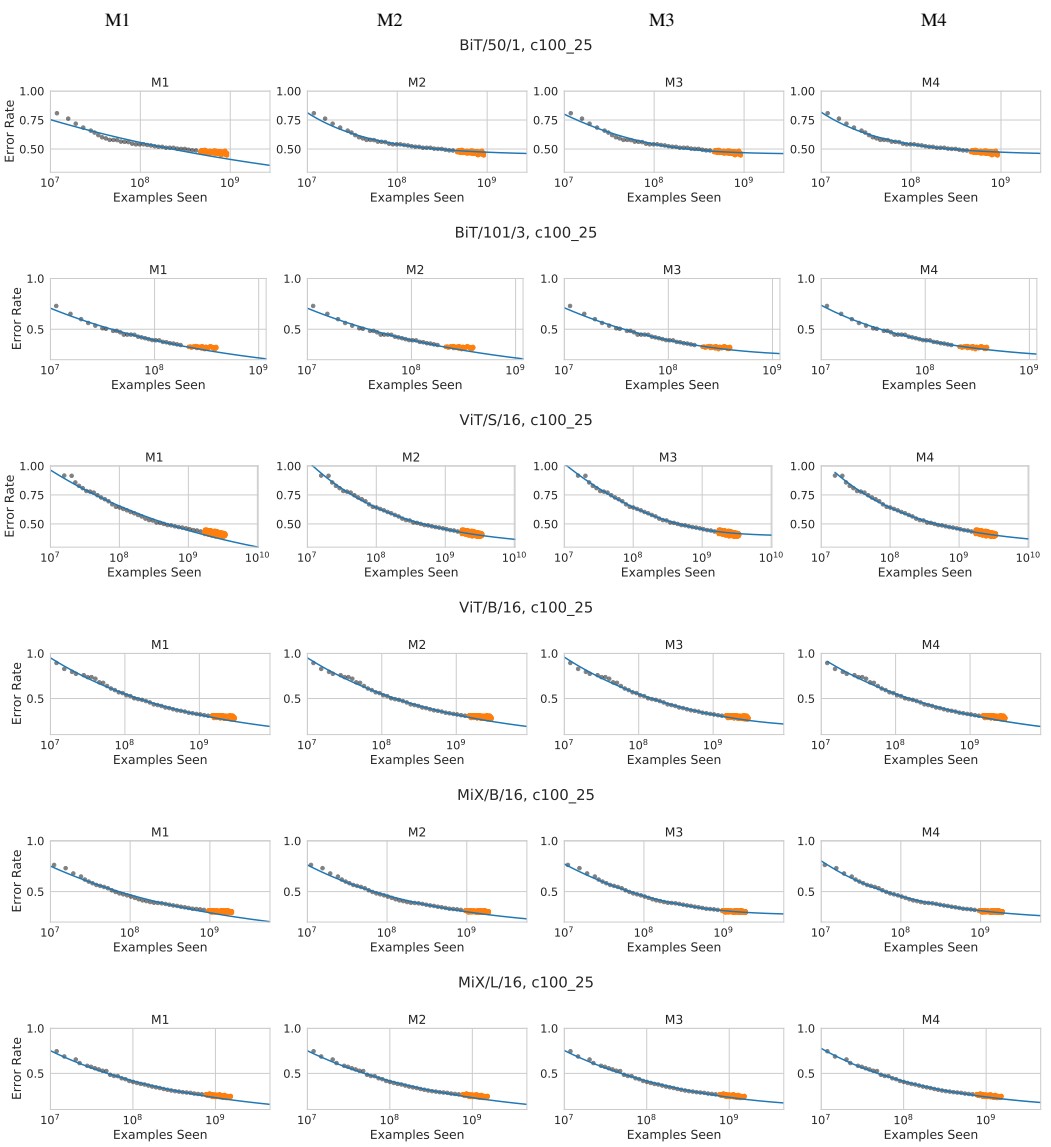

Figure 20: CIFAR100 25-shot.

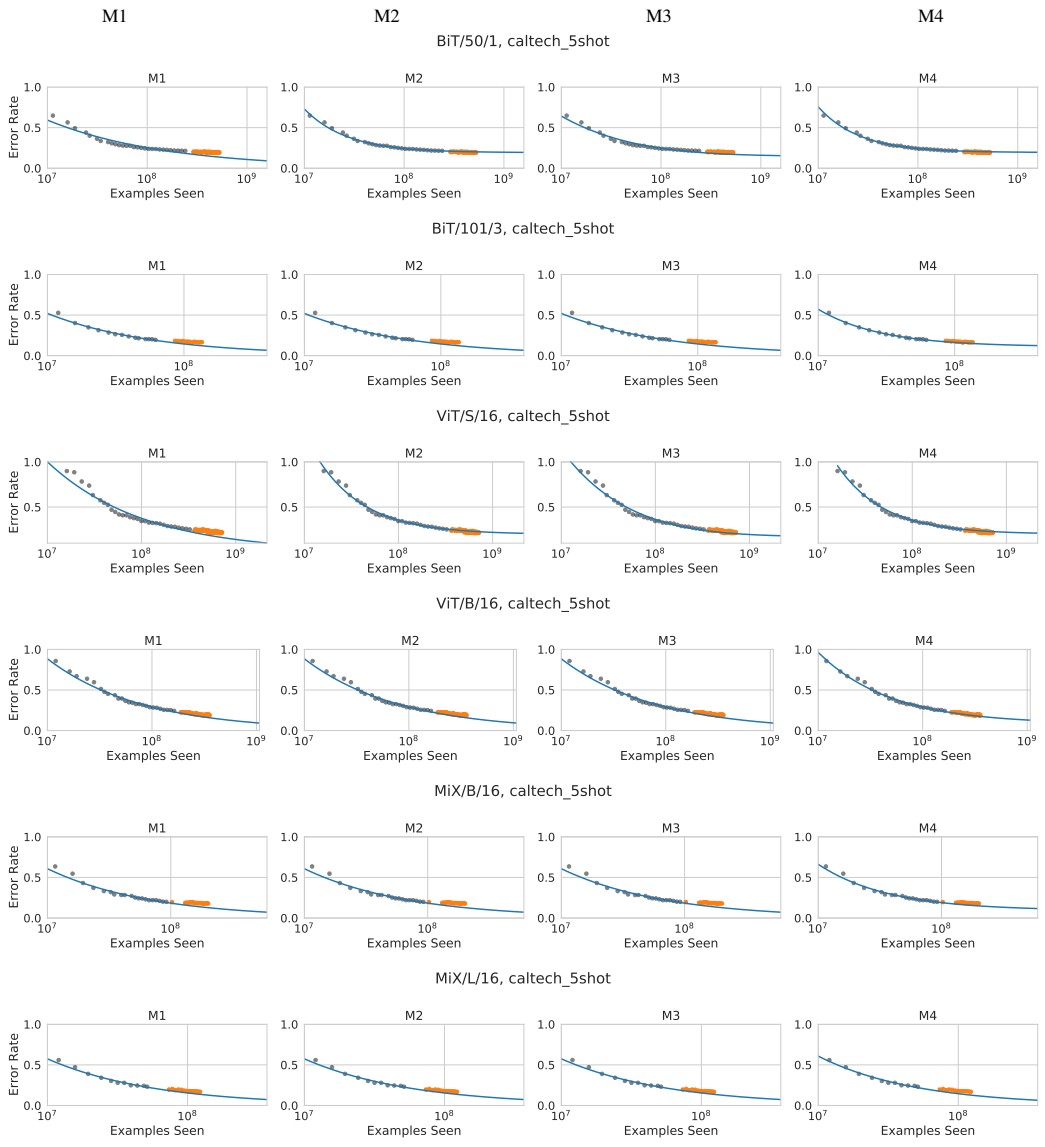

Figure 21: Caltech101 5-shot.

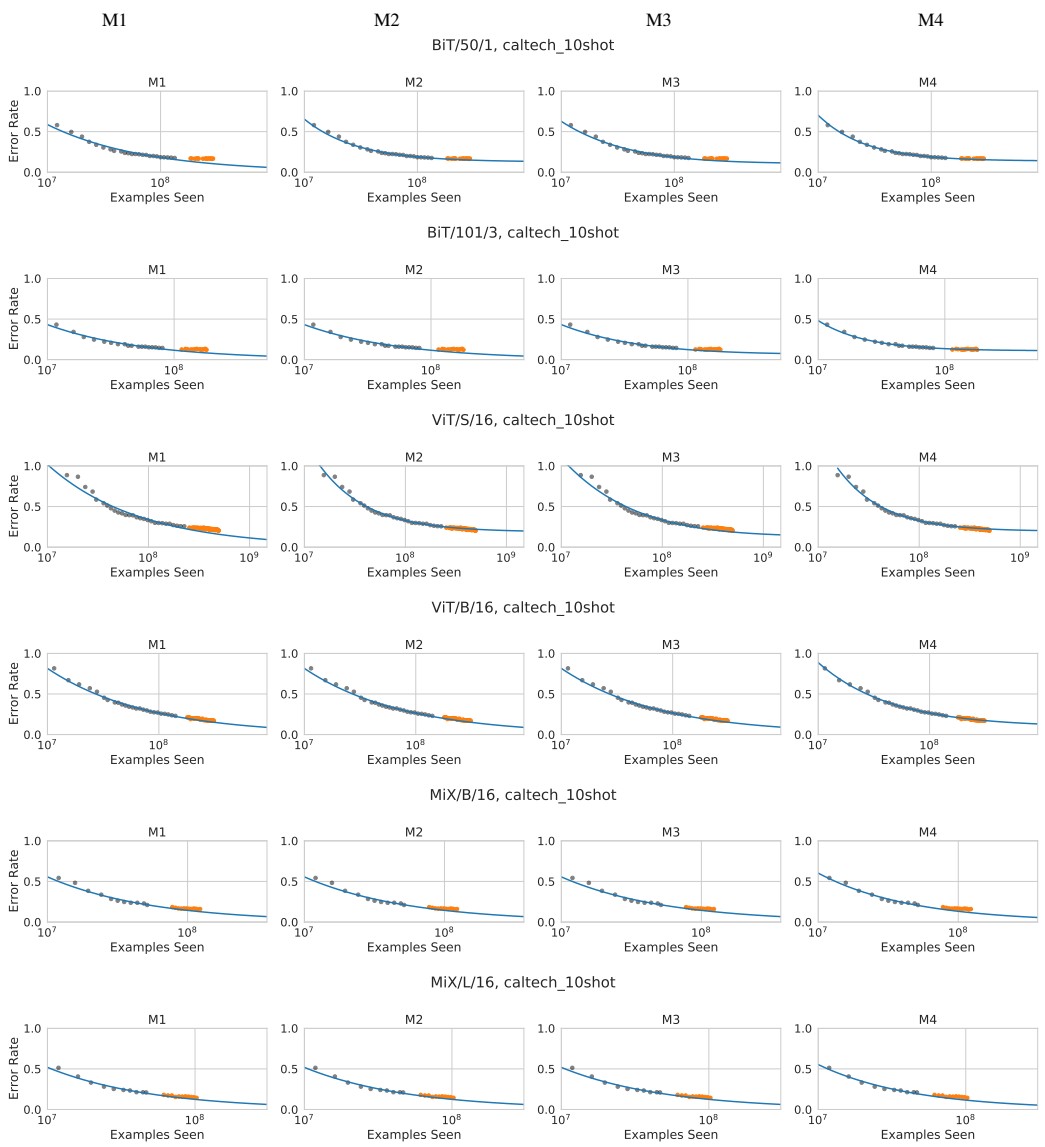

Figure 22: Caltech101 10-shot.

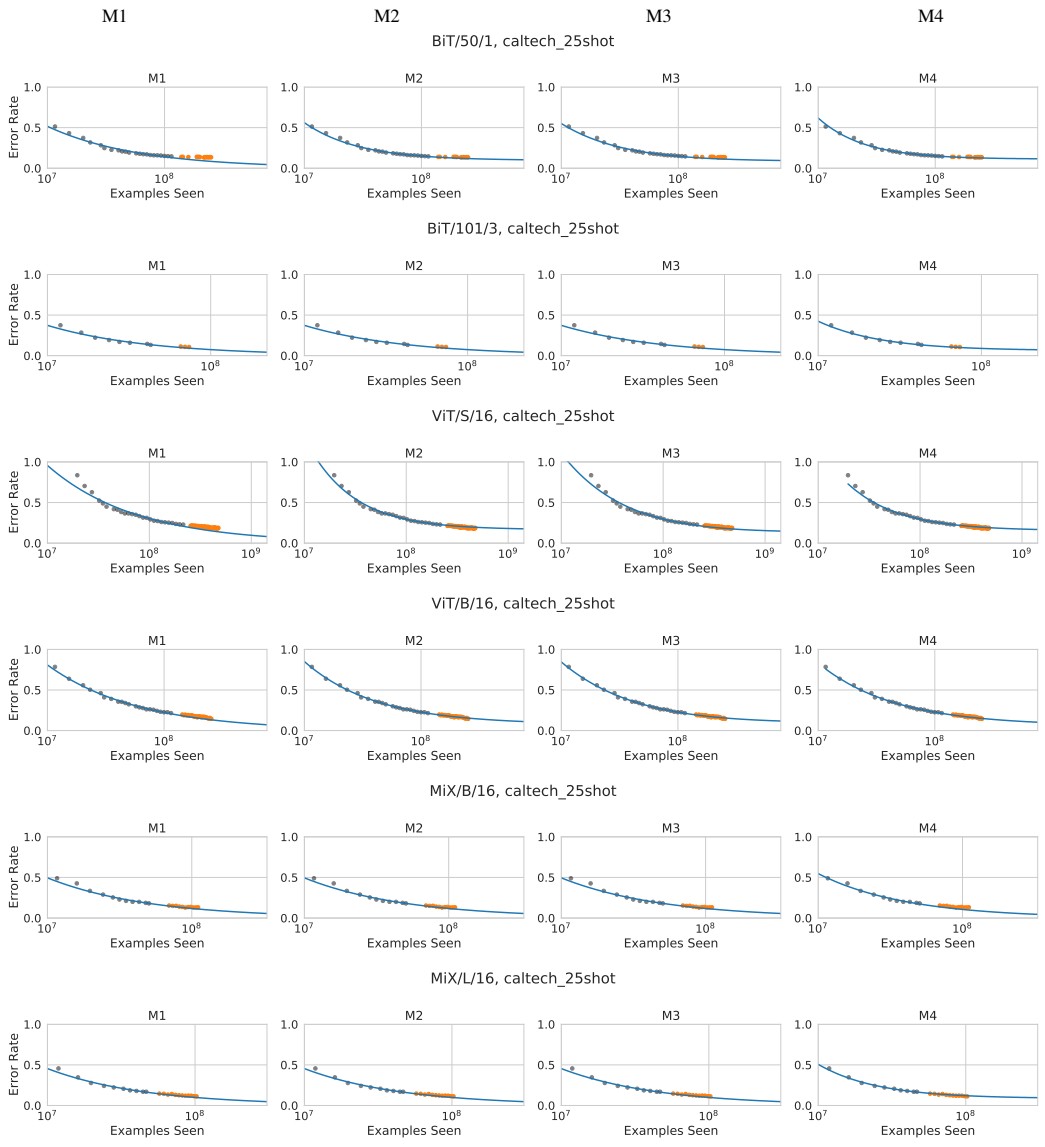

Figure 23: Caltech101 25-shot.