# OpenReview forum: "Revisiting Neural Scaling Laws in Language and Vision"
_NeurIPS.cc/2022/Conference — NeurIPS 2022 Accept_

### Official Review · Reviewer_am89 · 2022-07-08

**Rating:** 8
**Confidence:** 3
**Soundness:** 4 excellent
**Presentation:** 4 excellent
**Contribution:** 2 fair

**Summary:**

The authors propose an extrapolation-based fitting for better estimating the scaling properties of deep learning models (e.g., the larger model lowest Val/Test loss). The author compares interpolation methods, e.g.,  power-law fitting, with the proposed extrapolation method on different tasks image classification, NMT, language modeling, tasks from BIG-Bench, and architecture type and sizes. The results show that the proposed extrapolation method is more accurate in predicting the model's performance.

**Questions:**

- The loss shown in the appendix is quite important to highlight the difference between different methods. Why not include it in the main paper?

**Limitations:**

No this is missing in the paper.

**Strengths And Weaknesses:**

Strengths
- The experiments to support the claims are extensive and explore multiple modalities and architectures.
- The paper is well written, well-motivated, and easy to follow.
- Achieving better prediction in terms of model scaling (e.g., size, data size) is important in many scenarios, once for all neural architecture search.

---

> ### Author Response · Authors · 2022-08-01
> **Authors' Response**
>
> Thank you for your comments and suggestions. We appreciate the positive feedback on the thoroughness of the experimental evaluation, clarity of writing, and importance to the field.
>
> We will move the discussion in Appendix A.1 about the loss function used in estimating the scaling law parameters to the main paper in the camera ready version. We have kept it in the appendix due to the space limit. We agree that this is important for readability to highlight the differences between models.

---

### Official Review · Reviewer_frKW · 2022-07-10

**Rating:** 5
**Confidence:** 3
**Soundness:** 2 fair
**Presentation:** 2 fair
**Contribution:** 3 good

**Summary:**

This paper argues that predicting the benefit of scale should be conducted based on learning the learning curve 'extrapolation' rather than 'interpolation'. In addition, based on this, this work proposes an estimator model for the scaling laws, which extrapolates more accurately than previous methods, and in various experiments, the proposed estimator indeed shows better performance.

**Questions:**

- Isn’t it natural to derive the $\mathcal{M}_4$ based on the empirical observations rather than the synthetic experiment as in Section 3?
- I’m not sure why the first experiments were conducted in a few-shot setting instead of the pre-training task itself.
- In figure 3, does the x-axis, examples seen, means the number of data examples that are fed during training instead of the size of the dataset? Then, is it the realistic experimental setting where the interpolation region is set around $3\cdot10^8$ examples seen, which is the same as the number of data samples of JFT-300M?
- The observation that the $\mathcal{M}_4$ tends to yield estimates of the scaling exponents that are larger in absolute magnitude than in other methods means what?
- I’m confused that, for example in Figure 3, shouldn’t the difference in $c$ be reflected as different linear functions with different slopes rather than different curves in log-log plots?

**Limitations:**

This paper does not address the limitations and considerations about the negative societal impact.

**Strengths And Weaknesses:**

### Strengths

- I believe the task dealt with in this paper will be able to give a significant impact in the era where scaled-up DNN models show surprising performance.
- To support the claim of this work, extensive experiments with various models, datasets, and experimental environments are conducted.

### Weaknesses

- There are several concerns about the experiments and I listed the questions below.
- I guess the derivation of $\mathcal{M}_4$ arises from the motivation part, and the motivation part says an accurate model that extrapolates well would produce a linear curve, however, I was not able to get why it is true.
- It was a little confusing when I first read the paper, so I recommend providing the range of $\beta, c$ in the Introduction to help readers imagine the shape of the graphs.

---

> ### Author Response · Authors · 2022-08-01
> **Authors' Response**
>
> Thank you for your valuable feedback. We have updated the paper to address your concerns and provided responses to your questions. If your concerns are sufficiently addressed, we appreciate it if you would consider accepting our paper. Otherwise, please let us know about your remaining concerns.
>
> - *Linear curves in the motivation section*: In the synthetic experiment in Section 3, the Bayes risk $\delta$ is known *exactly*. We also know that the learning algorithm in the experiment is consistent so its error rate will approach $\delta$ as $x\to\infty$ (e.g. because we use a linear predictor, the optimal predictor is itself linear and the set of linear predictors has a finite VC dimension). So, $\varepsilon_x\to\delta$ as $x\to\infty$. In Figure 2, let $f(x)$ be the predictions of an estimator such as M2. We plot $f(x)-\delta$ against $x$. Since the error rate follows a power law asymptotically and $\delta$ is the ground-truth limit, an accurate predictor $f(x)$ in Figure 2 would produce a linear curve in a log-log plot for large $x$.
>
> - *Range of parameters*: Thank you for the suggestion. We have added this to the introduction (Lines 19 and 80).
>
> - *few-shot setting*: We have clarified this in the revised paper (Section 5.1). In image classification, pretraining is the common paradigm for scaling up models, where large models are pretrained on large data before fine-tuning on the downstream tasks. The end goal of pre-training is to better serve diverse downstream tasks. However, pre-training task performance is not representative of the downstream performance [1]. Thus, we use several representative downstream tasks (e.g. data-efficient classification on ImageNet) to perform the analysis, following [1].
>
> - *Figure 3*: The experiment in this figure is an illustration only to highlight how the scaling exponent that extrapolates best can be quite different from the scaling exponent that interpolates best. The interpolation data in this figure extends for up to 500M examples. We clarified this in the caption of Figure 3. We use “seen examples”, which are independent of the dataset size, where the former represents how many examples (including duplicates) seen during training while the latter represents how many unique examples are in the dataset. In the main evaluations in Section 5, we always use the same formula for the split in all cases as described in Lines 173-176.
>
> - *"The observation that the M4 tends to yield estimates of the scaling exponents that are larger in absolute magnitude than in other methods means what?"*: In Figure 6, M4 suggests that the models have more favorable scaling behavior than what would be predicted using M2. We clarified this in the caption of Figure 6.
>
> - *”I’m confused that, for example in Figure 3, shouldn’t the difference in c be reflected as different linear functions with different slopes rather than different curves in log-log plots?”*: This would be true if the limiting performance $\varepsilon_\infty$ is zero, i.e. where there is no saturation. However, the limiting error in ImageNet is likely bounded away from zero (see for example [2, 3]) so the curves saturate and they are not linear in a log-log plot.
>
> [1] Zhai, Xiaohua, et al. "Scaling vision transformers”, CVPR, 2022.
>
> [2] Beyer, Lucas, et al. "Are we done with imagenet?" arXiv:2006.07159, 2020.
>
> [3] Stock, Pierre, et al. "Convnets and Imagenet beyond accuracy: Understanding mistakes and uncovering biases." ECCV, 2018.

---

> > ### Comment · Reviewer_frKW · 2022-08-08
> > **Response to authors' comments**
> >
> > I would like to thank the authors for the detailed discussion and some of my questions are resolved.
> > Therefore, I will raise the score from 4 to 5.
> >
> > Also, I still wonder about the larger absolute magnitude and I want to ask your opinion about the meaning of having the larger absolute value. For instance, unlike the image classification, why do you think the absolute value of $c$ gets larger when the language model gets larger?

---

> > > ### Author Response · Authors · 2022-08-09
> > > **Thanks**
> > >
> > > Thank you for the constructive feedback. We interpret larger absolute magnitudes of the scaling exponents to imply faster convergence to the limit.

---

### Official Review · Reviewer_7qUq · 2022-07-11

**Rating:** 7
**Confidence:** 3
**Soundness:** 3 good
**Presentation:** 3 good
**Contribution:** 3 good

**Summary:**

The paper contributes a solid work with a new recipe to estimate scaling laws reliably from learning curves, which outperforms previous work on a variety of tasks. The paper also shows that in order to validate scaling law parameters, we should base on extrapolation of data points instead of interpolation of data points. The paper also contributes a benchmark dataset consists of 90 tasks, which I believe it is an important contribution to the community.



**Questions:**

(see the strengths and weakness)

**Strengths And Weaknesses:**

Pros:
- The paper is well-written for most parts.
- The new recipe is ingenious to me.
- The experiment section is solid. It covers a lot of different tasks with clear results supporting the benefits of using M4, the proposed method.
- Overall the paper contains materials that are very nice contribution to the field.

Cons:
- I don't think it is clear in the paper while the new recipe works that well compared to M2. Is it because M4 contains M2 as a special case or is it because M4 has a sigmoid like shape, which avoids failures of M2 as in Figure 2 left? I think this needs to be elaborate more.

- It is also not clear what exactly it means to be "extrapolation". In the introduction, the authors said that: ``Given the estimated scaling law parameters, one can then extrapolate by predicting the performance f(x) for large values of x". However, in the experiment section, it was explained that: "we divide the learning curve into two splits: (1) one split used for training the scaling law estimators, and one split used for evaluating extrapolation performance. We set the cutoff between the two splits to be equal to xmax/2, where xmax is the maximum value of x". With that, I am not sure the second split, which used for evaluating extrapolation performance reflects what was stated in the statement (i.e. large values of x).

Note: Equation 3 is the one that I could not follow. Perhaps a more explanation on this would help readers a lot.

Typo:

of which four seem to be more prominent -> four seem???

This is consistent with the fact that scaling exponents often satisfy c -> satisfies?

-------
Update: I wanted to thanks authors for answering my questions. I still have an issue with extrapolation and hopefully the authors will put some more efforts on explaining that in the paper. However, my overall impression is that the recipe is ingenious to me and it seems to work. So I just raised my score from 6 to 7.

---

> ### Author Response · Authors · 2022-08-01
> **Authors' Response**
>
> Thank you for your useful comments! We have added new results, revised the writing and provided the response below to address your comments. Please let us know if the revision prompted by your request has sufficiently improved the paper.
>
> - *Why does M4 work better than M2?*: We have added to the appendix new results that help answer this question; see Section A.2 including Figure 10 and Table 4. We believe that both of the conditions you mentioned are important. They were both used when deriving M4. We discuss this briefly in Lines 108-116. To support this claim, we have included in the revised paper Appendix A.2 on the performance in image classification when $\alpha$ is not introduced; so that M4 does not contain M2. Note that in the latter case, we continue to have sigmoid-like functions but the performance is poor compared to M4. On the other hand, since M2 performs worse than M4, having sigmoid-like functions is important as well.
>
> - *Extrapolation*: We have clarified this in the revised version of the paper (Lines 174-177). Given a collection of points $\{(x_0, y_0), … (x_n, y_n)\}$ where $x$ lies in some interval $[0, \tau]$, interpolation is finding a function $f(x)$ that can predict $y$ accurately when $x\in [0, \tau]$. Extrapolation is finding a function $f(x)$ that predicts $y$ accurately even when $x> \tau$.
> In our evaluation, we train the estimator on the domain $x\in [0, \tau]$ and evaluate it on the domain $x\in [\tau, 2\tau]$; e.g. in NMT, $\tau$ is 256M sentence pairs because the maximum number of sentence pairs used is 512M.
>
> - *Equation 3*: We have clarified this in the revised paper (Lines 129 - 131). Equation 3 is a second-order expansion of M4. We include it to show how M4 differs from M2. When $\alpha>0$, we observe that the excess loss $\varepsilon_x-\varepsilon_\infty$ predicted by M4 deviates from the power law when $x$ is small and the difference is $O(\alpha x^{2c})$ (suppressing other constants).

---

> > ### Comment · Reviewer_7qUq · 2022-08-08
> > **Thanks!**
> >
> > I wanted to thank authors for answering my questions. I still have an issue with extrapolation and hopefully the authors will put some more efforts on explaining that in the paper. However, my overall impression is that the recipe is ingenious to me and it seems to work. So I just raised my score from 6 to 7.

---

> > > ### Author Response · Authors · 2022-08-09
> > > **Thanks**
> > >
> > > Thank you for the constructive feedback. We will elaborate more in the camera-ready version.

---

### Author Response · Authors · 2022-08-01
**General Comments**

We thank the reviewers for the valuable comments. We appreciate their positive feedback on the novelty of the method, thoroughness of the experimental evaluation,  clarity of writing, and importance to the field.

We have improved the paper further by incorporating the suggestions made by the reviewers. Below is a summary of the changes:

- Added a new appendix A.2 to show that having sigmoid-like functions alone is not sufficient to have a small extrapolation loss, answering a question raised by Reviewer 7qUq. The new results also serve as an ablation for the parameter $\alpha$ in M4.
- Clarified the meaning of extrapolation in Lines 173-177 as suggested by Reviewer 7qUq.
- Elaborated more on Equation 3 in Lines 129-131 as suggested by Reviewer 7qUq.
- Included the range of parameters in Lines 19 and 80 as suggested by Reviewer frKW.
- Elaborated more on the motivation behind using few-shot evaluation in Image Classification in Section 5.1 as suggested by Reviewer frKW.
- Due to the space constraint, we kept the contents of appendix A.1 in the supplementary material and will move it to the main paper in the camera-ready version as suggested by Reviewer am89.

---

### Meta-Review · Area_Chair_tYTM · 2022-08-26

**Recommendation:** Accept
**Confidence:** Certain

**Metareview:**

The submission proposes a new method for deriving scaling laws for deep neural networks. Previous work has fit curves to datapoints based on interpolation. The paper argues convincingly that extrapolation is much more important - i.e. how well the function will predict future points if we trained longer. The authors derive a new approach to estimating scaling laws, and demonstrate that it outperforms existing proposals at predicting future points on a range of vision and language tasks. The paper notices a clear issue with current approaches to scaling laws, the intuition behind their approach is clear, and the experimental results are convincing - so I recommend acceptance.

**Award:**

No

---

### Decision · Program_Chairs · 2022-09-14

Accept